

# From ray tracing to waves of topological origin in continuous media

Antoine Venaille[1*], Yohei Onuki[1,2], Nicolas Perez[1] and Armand Leclerc[3]

**1** ENS de Lyon, CNRS, Laboratoire de Physique UMR5672, F-69342 Lyon, France
**2** Research Institute for Applied Mechanics, Kyushu University,
Kasuga, Fukuoka 816-8580, Japan
**3** ENS de Lyon, CNRS, Centre de Recherche Astrophysique de Lyon UMR5574,
F-69230, Saint-Genis Laval, France.

★ antoine.venaille@ens-lyon.fr

## Abstract

Continuous media commonly support a discrete number of wave modes that are trapped along interfaces defined by spatially varying parameters. In the case of multicomponent wave problems, those trapped modes fill a frequency gap between different wave bands. When they are robust against continuous deformations of parameters, such waves are said to be of topological origin. It has been realized over the last decades that waves of topological origin can be predicted by computing a single topological invariant, the first Chern number, in a dual bulk wave problem that is much simpler to solve than the original wave equation involving spatially varying coefficients. The correspondence between the simple bulk problem and the more complicated interface problem is usually justified by invoking an abstract index theorem. Here, by applying ray tracing machinery to the paradigmatic example of equatorial shallow water waves, we propose a physical interpretation of this correspondence. We first compute ray trajectories in the phase space given by position and wavenumber of the wave packet, using Wigner-Weyl transforms. We then apply a quantization condition to describe the spectral properties of the original wave operator. This bridges the gap between previous work by Littlejohn and Flynn showing manifestation of Berry curvature in ray tracing equations, and more recent studies that computed the Chern number of flow models by integrating the Berry curvature over a closed surface in parameter space. We find that an integral of Berry curvature over this closed surface emerges naturally from the quantization condition, which allows us to recover the bulk-interface correspondence.



## Contents

Owing to rotation, density stratification, compressibility, or magnetic fields, astrophysical and geophysical fluids support the propagation of waves at all scales. Those waves play a key role in redistributing energy or momentum in oceans, atmospheres, and stellar interiors. Such waves usually involve a coupling between several fields such as velocities in different directions and fluid density. Thus, astrophysical and geophysical waves are multicomponent wave problems. Textbook presentations of those waves usually start by linearizing a given flow model around a base state, and by computing the spectrum of a linear operator. This spectrum involves both a dispersion relation given by the eigenvalues of this linear operator and polarization relations between the different fields. the polarization relations are given by

eigenvectors of the linear operator. When the coefficients of the partial differential equations (PDE) are homogeneous (constant in space), solutions are easily found in unbounded geometries using a Fourier transform. Such *bulk wave problems* are simple to compute, and are useful to understand many aspects of wave propagation. For instance the computation of this bulk problem allows to classify waves in different wave bands such as sound waves, gravity waves, among others. Yet, most practical situations involve spatially varying parameters, which are in general intractable analytically.

There are, however, three powerful complementary theoretical approaches that can be used to tackle wave problems in nonuniform media with smooth variations: the study of simple but emblematic normal forms, the use of ray tracing equations, and the topological analysis:

- The first *normal form approach* is based on the idea that most salient features associated with spatial variations in the medium can be understood qualitatively by assuming linear variations of the parameters in one direction but constant in the other. A great success of this approach was the discovery of two unidirectional equatorially trapped waves by Matsuno in 1966 [1]. The drawback of this approach is its lack of generality, and there is no guarantee that the problem will be solvable analytically, although numerical computation of the spectrum remains always a possibility.

- The second *ray tracing approach* consists in computing the trajectories of wave packets, by assuming a scale separation between their wavelength and the spatial variations of the medium. This scale separation makes possible the use of standard asymptotic technique, such as the Wentzel-Kramers-Brillouin (WKB) approximation. This approach has long been used in fluids, and has even led to concrete applications in ocean dynamics [2,3], and atmospheric dynamics [4].

- The third *topological approach* is more recent. The main idea is that global features of the dispersion relation in inhomogeneous media can be classified and predicted. In particular, when a spatially varying parameter defines an interface, the wave spectrum may exhibit modes that are trapped along the interface, and that fill the frequency gap between different wave bands. Those modes are often referred to as *topological waves*, which is a shorthand term to mean *waves of topological origin*, when their presence is robust to continuous changes in parameters. It turns out that the emergence of such modes in *interface wave problems* can be predicted by computing a topological invariant for a set of *bulk wave problems that are much simpler to solve*. The topological invariant is an integer, the *Chern number*, that predicts the number of trapped modes in interface wave problems. Born in condensed matter, those ideas have irrigated all fields of physics, including now astrophysical and geophysical flows. For instance, a Chern number 2 was computed for rotating shallow water waves, consistently with the presence of two unidirectional modes in the Matsuno spectrum [5].

The topological methods may be thought of as a way to justify the outcomes of normal form approaches to understand more complicated situations. For instance, the computation of a topological invariant related to the Matsuno wave problem guarantees that the presence of two unidirectional modes trapped at the equator is a robust feature of equatorial waves, independently from the details of the planet's curvature, or from continuous changes of other parameters of the problem. In most previous applications of topology to geophysical and astrophysical flows, and more generally in continuous media, the correspondence between the topological invariant of the bulk problem and the number of topological waves (more precisely defined as a spectral flow later on) in the interface wave problem was justified by invoking an abstract index theorem [6–8]. The aim of these notes is to establish a connection between the ray tracing approach and the topological approach, in order to provide some physical intuition

on this *bulk-interface correspondence*, building upon previous works on equatorial waves.

More precisely, we intend to relate the seminal work of Littlejohn and Flynn [9] on ray tracing equations for multicomponent wave systems, to more recent works on topological waves in continuous media, spectral flows and Berry monopole as presented for instance in the lecture notes of Faure [7] or Delplace [8]. In both cases, a fundamental role is played by the Berry curvature, which can be computed from the knowledge of the bulk wave polarization relations. It was shown that ray trajectories in position/wavenumber phase space are modified by this term just as a charged particle is deviated by a magnetic field [9]. This has found application from condensed matter [10–12] to geophysical fluid dynamics [13].

In the context of topological waves, the same Berry curvature (a geometrical quantity) is used to compute the Chern number (a topological quantity). This Chern number counts singularities in an abstract space of bulk wave polarization relations parameterized over a closed surface. The Chern number is related to the Berry curvature through a generalization of the Gauss-Bonnet formula that relates the genus of a surface to the integral of the Gaussian curvature over this surface [7, 8]. It is thus tempting to establish a connection between ray tracing, which describes the dynamics of a local wave packet influenced by the Berry curvature, to the global spectrum of a wave operator with spatially varying coefficient, which is ruled by a single Chern number.

The duality between ray tracing and spectral properties of an operator is reminiscent of the duality between classical and quantum mechanics. Classical mechanics deals with ordinary differential equations (ODE) that describe trajectories in phase space, while quantum mechanics deals with PDE that describe wave functions. Originally, quantum mechanics was derived by proposing quantization procedures that map functions of phase space variables such as the classical Hamiltonian to operators of quantum mechanics such as the Schrödinger operator [14]. Semi-classical analysis is the reverse procedure that derives classical mechanics trajectories in phase space (and possible corrections) in the limit of vanishing Planck constant, as done for instance by using the celebrated WKB ansatz [15]. Those techniques are not restricted to quantum mechanics, and a branch of mathematics called microlocal analysis was developed in the seventies to establish connections between ray tracing and spectral properties of operators, see e.g. [16].

The connection between ray tracing and spectral properties of an operator makes use of the Wigner-Weyl transform, that relates in a systematic way a wave operator to its symbol, which is a function playing for instance the role of the Hamiltonian in classical mechanics. Such tools have been used for instance recently by mathematicians to explain the emergence of singular wave patterns in density-stratified flows [17], or the dynamics of equatorial waves [18–20], including through the use of topological tools [7]. This Weyl-Wigner formalism has also been used by theoretical physicists to describe wave transport phenomena in continuous media [9], including geophysical flows [21, 22], see also the recent work by Onuki for a pedagogical introduction to the formalism in the context of geophysical flows [23]. Our contribution will be to connect those works with previous studies on topological waves in continuous media such as geophysical flows [5, 24], astrophysical flows [25, 26], or plasmas [27, 28].

We present in section 1 the equatorial shallow water model [1], and use this example to introduce the notion of spectral flow index [7, 8], which is an integer that counts the number of topological modes that transit from one wave band to another in the spectrum of a wave operator, when a parameter is varied from $-\infty$ to $+\infty$. Those two limiting cases allow us to define the equivalent of a semi-classical limit.

We review in section 2 the standard procedure to diagonalize a multicomponent wave operator, taking advantage of a small parameter in the problem in the semi-classical limit, building

upon [9, 12, 23]. This derivation makes use of the Wigner-Weyl transform and notions of symbolic calculus that are presented in Appendix A. The diagonalization of the multicomponent wave operator allows us to define a scalar operator describing the dynamics of a wave packet in a given wave band.

We derive in section 3 the dynamics of those wave packets, which allows us to describe ray trajectories in the phase space defined by position and momentum of the wave packets. Those results were first derived by [9], who highlighted the central role of the Berry curvature, and a duality between two different possible definitions of position/wavenumber of the wave packets, one of them being independent from any gauge choice done during the diagonalization of the original wave operator. Our contribution is to propose a new physical interpretation of the transformation between two different phase space coordinates for the wave packets, and to explain that the Berry curvature appearing in the ray tracing equations is related to the presence of a topological invariant in the bulk wave problem: the first Chern number.

We recover in section 4 the spectral properties of the wave operator by applying a quantization condition (imposing that the phase picked up by a wave packet along a trajectory is a multiple of $2\pi$), as proposed in [9]. Our contribution is to show how this quantization can be used to recover the spectral flow result noticed in section 1. The derivation of the quantization condition will highlight terms than can be interpreted as integral of the Berry curvature over a closed surface, which is nothing but the Chern number, a topological invariant. Thus, we show how spectral properties are related to topological properties through ray tracing. We finally propose an alternative interpretation of the spectral flow result based on the modification of the phase space density of states by the Berry curvature, a standard result in non-canonical Hamiltonian systems.

**Notations** Any operator will involve a hat, as $\hat{\Omega}$. Operators or functions that are underlined once means that they involve multiple components. Operators underlined twice are matrices of operators. A character underlined twice but without a hat is a matrix of functions. Bold symbols such as $\mathbf{z}$ are vectors in phase space. A tilde is placed on a dimensional variable and it will be removed to define a scaled dimensionless one.

# 1 Motivation from equatorial shallow water waves

## 1.1 Wave equation and definition of the spectral flow

**The shallow water model** describes the dynamics of a thin layer of fluid with homogeneous density under gravity (constant $g$), with a solid boundary at the bottom and a free surface at the top [29]. For the sake of simplicity, we assume that the dynamics takes place in a plane tangent to Earth near the equator, over a flat bottom, and the fluid layer thickness is denoted by $H$ (see Fig. 1).

In this geometry, the dynamics is conveniently expressed in Cartesian coordinates $(x, y)$, where $x$ points eastward (zonal direction) and $y$ points northward (meridional direction). Metric terms are neglected, but planet's sphericity is taken into account by considering linear variations of the Coriolis parameter in the meridional direction $y$. This Coriolis parameter denoted by $\beta y$ is twice the projection of the planet's rotation vector on the local vertical axis, as in the Foucault pendulum experiment. The linearized shallow water dynamics around a state of rest reads

$$\frac{\partial}{\partial \tilde{t}} \begin{pmatrix} \tilde{u} \\ \tilde{v} \\ \tilde{\eta} \end{pmatrix} = \begin{pmatrix} 0 & \beta \tilde{y} & -g\frac{\partial}{\partial \tilde{x}} \\ -\beta \tilde{y} & 0 & -g\frac{\partial}{\partial \tilde{y}} \\ -H\frac{\partial}{\partial \tilde{x}} & -H\frac{\partial}{\partial \tilde{y}} & 0 \end{pmatrix} \begin{pmatrix} \tilde{u} \\ \tilde{v} \\ \tilde{\eta} \end{pmatrix}, \tag{1}$$

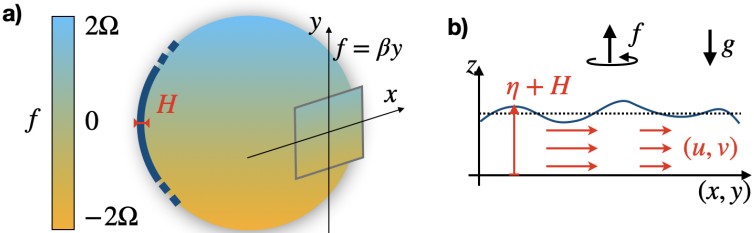

Figure 1: **a)** The Coriolis parameter $f$ as a function of latitude on a rotating planet, and the shallow water model with $H$ the fluid layer thickness, much smaller than the typical horizontal scales of motion. **b)** Beta-plane approximation: the flow takes place in a plane $(x, y)$ tangent to the equator, with $f = \beta y$.

where $(u, v)$ is a depth-independent two-dimensional velocity field and $\eta$ is the variation of surface elevation with respect to the rest state.[1]

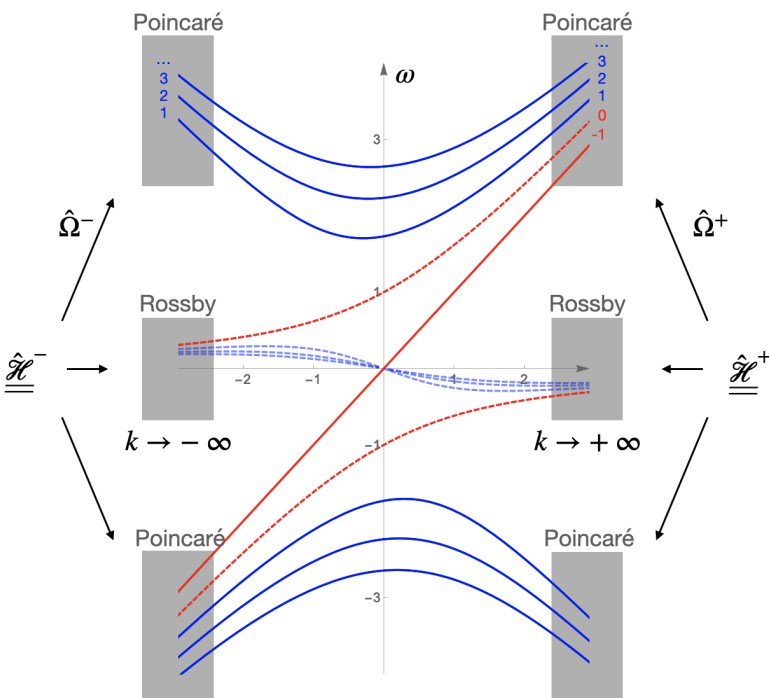

Figure 2: Eigenvalues of the beta plane shallow water wave operator defined in (3), as a function of the zonal wavenumber $k$, for $\beta = 1, c = 1$, often referred to as the Matsuno spectrum [1]. Two modes (in red) are gained by the positive-frequency Poincaré wave band as $k$ increases. The aim of this paper is to interpret this spectral flow using semi-classical analysis.

**A one-dimensional wave problem** can be derived from this model. We take advantage of the invariance with $x$ to write

$$(\tilde{u}, \tilde{v}, \tilde{\eta}) = (cu, cv, H\eta)\, e^{ik\tilde{x}}, \quad c = \sqrt{gH}. \tag{2}$$

---

[1]Taking into account spatial variations of bottom elevation involves additional terms in the last row of the wave operator (1). Those additional terms lead to interesting geometrical and topological properties [23, 24].

From now on we chose $\beta = 1$ and $c = 1$, which can always be done by adimensionalizing time and length units with $1/\sqrt{\beta c}$ and $\sqrt{c/\beta}$, respectively. The wave problem is now

$$i\frac{\partial}{\partial \tilde{t}}\begin{pmatrix} u \\ v \\ \eta \end{pmatrix} = \underline{\underline{\hat{\mathcal{H}}}}_k \begin{pmatrix} u \\ v \\ \eta \end{pmatrix}, \quad \underline{\underline{\hat{\mathcal{H}}}}_k = \begin{pmatrix} 0 & i\tilde{y} & k \\ -i\tilde{y} & 0 & -i\frac{\partial}{\partial \tilde{y}} \\ k & -i\frac{\partial}{\partial \tilde{y}} & 0 \end{pmatrix}. \tag{3}$$

The operator $\underline{\underline{\hat{\mathcal{H}}}}_k$ depends on a single parameter $k$. This problem was fully solved by Matsuno, and the corresponding dispersion relation is displayed in Fig. (2). One can look at how the wave spectrum changes when this parameter is varied:

*When varying the parameter $k$ from $-\infty$ to $+\infty$, two modes are gained by the positive-frequency wave band, called the Poincaré wave band.*

Similarly, two modes are lost by the negative frequency Poincaré wave band. By contrast, the central wave band called the Rossby wave band has a net gain of modes equal to 0: there are as many branches that leave the Rossby wave band as branches that enter the Rossby wave band when $k$ is varied from $-\infty$ to $+\infty$.

The bottom line is that some of the branches in the dispersion relation transit from one wave band to another when $k$ is varied, which defines a *spectral flow*,[2] and $k$ is called *the spectral flow parameter*. The spectral flow is quantified by an integer, the *spectral flow index*, that counts how many modes (branches in the dispersion relation) are gained or lost as $k$ is varied from $-\infty$ to $+\infty$. More precise and rigorous definitions of this index are given for instance in [7], and its relation with another integer named *analytical index* is explained in [8].

Matsuno's computation leads to a spectral flow index $+2$ associated with positive-frequency Poincaré wave band, 0 for the Rossby wave band, and $-2$ for the negative frequency Poincaré wave band.

Note that in many cases, the spectral flow is directly related to the difference between the number of right-moving (positive group velocity) and left-moving modes in a given range of frequencies, most often taken within the gap between different bulk wavebands when it exists. In Matsuno's case, at any nonzero frequency, there are two more modes with eastward group velocity than modes with westward group velocity, and this is related to the spectral flow index $\pm 2$ of the upper and low wave bands [5]. Thus, since group velocity corresponds to energy transport, spectral flow is related to unidirectional wave transport in that case.

## 1.2 The semi-classical limit for equatorial shallow water waves

Our aim is to understand this spectral flow from a semi-classical perspective, which is related to the familiar WKB approximation commonly used in geophysical fluid dynamics. For that purpose we take advantage of the existence of a small parameter in the limit $k \to \pm\infty$, by rescaling the wave equation as follows:

$$(\tilde{t}, \tilde{y}) = |k|(t, y), \quad \epsilon = \frac{1}{k^2}, \tag{4}$$

which leads to

$$i\epsilon\frac{\partial}{\partial t}\underline{\Psi} = \underline{\underline{\hat{\mathcal{H}}}}^{\pm}\underline{\Psi}, \quad \underline{\underline{\hat{\mathcal{H}}}}^{\pm} = \begin{pmatrix} 0 & iy & \pm 1 \\ -iy & 0 & -i\epsilon\frac{\partial}{\partial y} \\ \pm 1 & -i\epsilon\frac{\partial}{\partial y} & 0 \end{pmatrix}, \quad \underline{\Psi} = \frac{1}{\sqrt{2}}\begin{pmatrix} u \\ v \\ \eta \end{pmatrix}. \tag{5}$$

---

[2]The term spectral flow is unrelated to any actual flow in physical space.

The parameter $\epsilon$ appears only in front of the $y$-derivative, which is suggestive of the reduced Planck constant $\hbar$ in quantum mechanics. This is why the limit $\epsilon \to 0$ will from now on be referred to as a *semi-classical limit*, and $\epsilon$ will be called the *semi-classical parameter*.[3] Given that the length unit used to adimensionalize equations is $\sqrt{c/\beta}$, the semi-classical parameter compares this intrinsic length scale, called the equatorial radius of deformation, to the zonal wavelength $1/k$. Since $\sqrt{c/\beta}$ corresponds to a trapping length scale in the meridional direction, the semi-classical limit is a limit of large meridional to zonal aspect ratio for the waves. The upper-script index $\pm$ is used to distinguish the limit $k \to +\infty$ from the limit $k \to -\infty$, which both correspond to the semi-classical scaling $\epsilon \to 0$. An important remark follows:

*Understanding the spectral flow can be tackled by comparing the spectral properties of the operators $\hat{\underline{\underline{\mathcal{H}}}}^{+}$ and $\hat{\underline{\underline{\mathcal{H}}}}^{-}$ in the semi-classical limit $\epsilon \to 0$. More precisely, we will pair together modes of the two operators that share common properties. By continuity of the eigenvalues, the modes that can not be paired will be those belonging to a branch that transits from one wave band to another as $k$ varies.*

## 1.3 Strategy to interpret the spectral flow

We will show in section 2 how to project the original multi-component wave operator $\hat{\underline{\underline{\mathcal{H}}}}^{\pm}$ into scalar operators for each wave band, denoted by $\hat{\Omega}^{(j)\pm}$, where $j$ is the band index. Those scalar operators are different for the three wave bands, just as their dispersion relation are different. In fact, we will highlight a formal duality between such operators and the corresponding dispersion relations.

For that purpose, we will follow [9] and [23] and rely on the Wigner-Weyl transform that is introduced in Appendix A.1. After this first step, the spectral flow index for each wave band will be understood loosely as the difference between two infinite numbers

$$\mathcal{N}^{(j)} = \# \text{ of modes for } \hat{\Omega}^{(j)+} - \# \text{ of modes for } \hat{\Omega}^{(j)-}, \tag{6}$$

where $\#$ means "number".

We will derive in section 3 the ray tracing equation in wavenumber/physical space (phase space), for wave packets described by operators $\hat{\Omega}^{(j)\pm}$. We will highlight the central role played by a quantity called the Berry curvature in ray tracing equations, as noted first in [9]. We will explain how to compute this Berry curvature from the knowledge of the wave polarization relations. Using an integration over a surface in parameter space, we will relate this curvature, which is a local geometrical quantity, to a singularity described by a topological invariant named the first Chern number, which describes global properties of continuous families of eigenvectors.

We will show in section 4 how to recover the spectrum of $\hat{\Omega}^{(j)\pm}$ from the use of a quantization procedure of ray trajectories in phase space, following [9]. The quantization procedure amounts to assuming that the phase picked-up by a wave packet along a trajectory is an integer multiple of $2\pi$. This is analogous to the Bohr-Sommerfeld quantization in quantum mechanics. This procedure will make possible an explicit computation of the mode imbalance (6). This will also highlight the topological origin of this mode imbalance, that is related to an integral of the Berry curvature over a closed surface, and thus to the Chern number.

---

[3]The name WKB parameter can also be used, but this terminology is sometimes kept for a specific class of scalar equations.

The index ($j$) for the wave bands will be dropped in the following when discussing general properties of the operator $\hat{\Omega}^{(j)\pm}$. In many steps of the derivations, the formula will be sufficiently general to apply for a variety of flow models. However, for pedagogical reasons we chose to focus on the particular case of shallow water waves.

## 2 Scalar wave operator and symbols

### 2.1 From multicomponent to scalar wave operator

The initial multicomponent wave problem is written formally as

$$i\epsilon\partial_t\underline{\Psi} = \underline{\hat{\mathcal{H}}}\,\underline{\Psi}\,, \tag{7}$$

where $\underline{\Psi}$ is a multicomponent wave field and $\underline{\hat{\mathcal{H}}}$ a linear operator involving spatially varying coefficients and spatial derivatives, together with an external parameter $k$ to be varied. In the shallow water case, this *spectral flow parameter* $k$ is the wavenumber in zonal (West-East) direction. Here we will focus only on 1D problems involving the spatial (meridional) coordinate $y$, with the corresponding derivative $\partial_y$. The parameter to be varied will be the wavenumber $k$ in the $x$ (zonal) direction. An example is given in the previous section, where $\underline{\hat{\mathcal{H}}}$ is to be replaced by $\underline{\hat{\mathcal{H}}}^{\pm}$.

We recall here general results that are presented in a pedagogical way by Onuki [23] (for fluid dynamicists) and Reijnders *et al* [12] (for condensed matter physicists). We would like to express the multicomponent wave dynamics on the form of a scalar equation for a single wavefield denoted by $\psi$, assumed to be written on the following form:

$$i\epsilon\partial_t\psi = \hat{\Omega}\psi\,. \tag{8}$$

We also assume that reconstruction of the multicomponent wavefield from the scalar field is formally expressed with a multicomponent operator $\underline{\hat{\chi}}$ (a vector whose each component is an operator depending on the $y$ coordinate):

$$\underline{\Psi} = \underline{\hat{\chi}}\psi\,, \quad \underline{\hat{\chi}}^{\dagger}\cdot\underline{\hat{\chi}} = \mathbb{1}\,, \tag{9}$$

where $\mathbb{1}$ is the identity operator. The second equality guarantees that both $\underline{\Psi}$ and $\psi$ are normalized:

$$\int \mathrm{d}y\ \underline{\Psi}^{\dagger}\cdot\underline{\Psi} = \int \mathrm{d}y\ \psi^*\psi = 1\,. \tag{10}$$

In the case of shallow water waves, this norm represents the total energy of the flow,[4] which is the sum of kinetic plus available potential energy.

At this stage the operators $\hat{\Omega}$ and $\underline{\hat{\chi}}$ are not known. A useful equation relating those operators is obtained by combining (7), (8), and (9):

$$\underline{\hat{\mathcal{H}}}\,\underline{\hat{\chi}} = \underline{\hat{\chi}}\hat{\Omega}\,. \tag{11}$$

We will see in the next section that the expression for $\hat{\Omega}$ and $\underline{\hat{\chi}}$ can be obtained in the semiclassical limit, and are different for each wave bands of the original multi-component wave problem.

---

[4]The total energy is conserved, hence the normalization of the wave fields.

## 2.2 Expression of the scalar operator in the semi-classical limit

Our aim here is to make an educated use of Wigner transform, symbolic calculus, and Weyl transform to find the expression of the scalar operator $\hat{\Omega}$, by exploiting (11). Those tools are introduced in Appendix A. If an operator such as $\hat{\Omega}$ is known, then the Wigner transform defined by (A.2) yields its symbol $\Omega(y, p)$. This symbol is a function of $y$ and $p$, the conjugate momentum in the direction $y$, which is a scaled wavenumber. This procedure generalizes the Fourier transform to problems with spatially varying coefficients. As such, the symbol $\Omega(y, p)$ can be interpreted as a local dispersion relation for a plane wave oscillating much faster than the spatial variations of the parameter in the medium. The inverse transform that builds an operator $\hat{\Omega}$ from the knowledge of a function $\Omega(y, p)$ is called the Weyl transform, and is defined in Appendix A.1, equation (A.3). The operator $\hat{\Omega}$ is called a pseudo-differential operator as it cannot always be expressed as an explicit polynomial in $\partial_y$ [30].

Formally, all the operators in (11) can be expanded as

$$\hat{\underline{\underline{\mathcal{H}}}} = \hat{\underline{\underline{\mathcal{H}}}}_0 + \epsilon \hat{\underline{\underline{\mathcal{H}}}}_1 + \mathcal{O}(\epsilon^2), \tag{12}$$

$$\hat{\underline{\chi}} = \hat{\underline{\chi}}_0 + \epsilon \hat{\underline{\chi}}_1 + \mathcal{O}(\epsilon^2), \tag{13}$$

$$\hat{\Omega} = \hat{\Omega}_0 + \epsilon \hat{\Omega}_1 + \mathcal{O}(\epsilon^2). \tag{14}$$

In the following, removing the hat (ˆ) from an operator will refer to its symbol. Symbols can also be expanded in the semi-classical limit as

$$\underline{\underline{\mathcal{H}}} = \underline{\underline{\mathcal{H}}}_0 + \epsilon \underline{\underline{\mathcal{H}}}_1 + \mathcal{O}(\epsilon^2), \tag{15}$$

$$\underline{\chi} = \underline{\chi}_0 + \epsilon \underline{\chi}_1 + \mathcal{O}(\epsilon^2), \tag{16}$$

$$\Omega = \Omega_0 + \epsilon \Omega_1 + \mathcal{O}(\epsilon^2). \tag{17}$$

Products of operators $\hat{a}\hat{b}$ are in general different from the operator $\widehat{ab}$ obtained by a Weyl transform of the standard product between symbols $a(y, p)$ and $b(y, p)$. As explained in Appendix A.3, equation (A.15), we can however define a particular symbol product denoted by $a \star b$ such that the Weyl transform of this star product corresponds to the standard product of operators (i.e. their composition). Let us apply this procedure to develop the operator products in (11). The r.h.s. is expanded as

$$\hat{\underline{\chi}}\hat{\Omega} = \widehat{\underline{\chi} \star \Omega}. \tag{18}$$

The star product is computed by using its definition (A.15) together with the symbol expansions (16) and (17) in the semi-classical limit, which yields

$$\underline{\chi} \star \Omega = \underline{\chi}_0 \Omega_0 + \epsilon \left( \underline{\chi}_1 \Omega_0 + \underline{\chi}_0 \Omega_1 + \frac{i}{2} \left\{ \underline{\chi}_0, \Omega_0 \right\} \right) + \mathcal{O}(\epsilon^2), \tag{19}$$

where we have introduced the Poisson bracket for two symbols $\underline{\chi}_0(y, p)$ and $\Omega_0(y, p)$:

$$\left\{ \underline{\chi}_0, \Omega_0 \right\} = \partial_y \underline{\chi}_0 \partial_p \Omega_0 - \partial_p \underline{\chi}_0 \partial_y \Omega_0. \tag{20}$$

The corresponding operator is finally obtained by applying the Weyl transform (A.3) to the star product.

## 2.3 Asymptotic expansion up to order one

Using symbolic calculus and the asymptotic expansion introduced in section 2.2, one can now exploit the equality (11) to find expressions of $\underline{\chi}_0$, $\Omega_0$ and $\Omega_1$. At the leading order, we obtain

a matrix eigenvalue equation for $\underline{\underline{\mathcal{H}}}_0$ :

$$\underline{\underline{\mathcal{H}}}_0 \underline{\chi}_0 = \Omega_0 \underline{\chi}_0 , \quad \underline{\chi}_0^\dagger \underline{\chi}_0 = 1 . \tag{21}$$

We see that the symbols $\Omega_0$ and $\underline{\chi}_0$ are just the eigenvalue and the eigenvector of $\underline{\underline{\mathcal{H}}}_0$, the leading order component of the symbol associated with the multicomponent wave operator. This zeroth order result may be understood as the outcome of a "local" plane wave solution: in this framework, $\underline{\chi}_0$ is the local polarization vector associated with the local dispersion relation $\Omega_0(y, p)$.

Collecting the first order terms in (11), multiplying on the left by $\underline{\chi}_0^\dagger$, and using $\underline{\chi}_0^\dagger \underline{\underline{\mathcal{H}}}_0 = \underline{\chi}_0^\dagger \Omega_0$ (owing to the hermiticity of the wave operator[5]) leads to

$$\Omega_1 = \Omega_{1A} + \Omega_{1B} , \tag{22}$$

$$\Omega_{1A} = \underline{\chi}_0^\dagger \underline{\underline{\mathcal{H}}}_1 \underline{\chi}_0 + \frac{i}{2} \underline{\chi}_0^\dagger \left\{ \underline{\underline{\mathcal{H}}}_0 , \underline{\chi}_0 \right\} + \frac{i}{2} \underline{\chi}_0^\dagger \left\{ \underline{\chi}_0 , \Omega_0 \right\} , \tag{23}$$

$$\Omega_{1B} = -i \underline{\chi}_0^\dagger \left\{ \underline{\chi}_0 , \Omega_0 \right\} . \tag{24}$$

More detailed on this computation can be found for instance in [9, 12].

Let us now explain why we have separated the first order expression into two components $\Omega_{1A}$ and $\Omega_{1B}$. The important point noticed by Littlejohn and Flynn is that the eigenvectors $\underline{\chi}_0$ of the zeroth order equation are defined up to a phase factor [9]. In physical jargon, choosing this phase amounts to a gauge choice. It appears that the first order expression of the scalar symbol (and the corresponding operator $\hat{\Omega}_1$) depends on this gauge choice. It is however possible to split the symbol into a part $\Omega_{1A}$ that is gauge independent and a part $\Omega_{1B}$ that is not gauge independent. This can be checked by applying the transformation $\underline{\chi}_0 \to \underline{\chi}_0 e^{ig(y,p)}$, where $g(y, p)$ an arbitrary real-valued function. The term $\Omega_{1A}$ is left unchanged, while the term $\Omega_{1B}$ is shifted as $\Omega_{1B} \to \Omega_{1B} + \{g, \Omega_0\}$. As we shall see later, the term $\Omega_{1B}$ is related to the Berry curvature describing local variations of the eigenvectors $\underline{\chi}_0$ in parameter space (i.e. local variations of the polarization relation in a given wave band). As such, the term $\Omega_{1B}$ is sometimes called the Berry term.

## 2.4 Application to the shallow water wave problem

Let us come back to the shallow water wave problem introduced in (5). The symbol of the wave operator $\hat{\underline{\underline{\mathcal{H}}}}^\pm$ is of order 0 in $\epsilon$:

$$\underline{\underline{\mathcal{H}}}_0^\pm = \begin{pmatrix} 0 & iy & \pm 1 \\ -iy & 0 & p \\ \pm 1 & p & 0 \end{pmatrix} , \quad \underline{\underline{\mathcal{H}}}_1^\pm = 0 . \tag{25}$$

The three eigenvalues of the symbol are, at leading order:

$$\Omega_0^\pm = \{-\omega_0, \, 0, \, \omega_0\} , \quad \text{with} \quad \omega_0 = \sqrt{y^2 + p^2 + 1} . \tag{26}$$

Notice that the eigenvalues of $\underline{\underline{\mathcal{H}}}_0^+$ and $\underline{\underline{\mathcal{H}}}_0^-$ are the same. The difference between the spectra of both operators manifests at next order (equations (23)-(24)), for which the expression of

---

[5]This is the only step involving the Hermicity assumption, which allows to cancel some of the terms. The diagonalization prodecure does not rely on this assumption, and could be performed, albeit with additional terms.

the eigenvectors at zeroth order is needed:

$$
\underline{\chi}_0^\pm = \left\{ \frac{1}{\sqrt{2}\sqrt{1+p^2}} \begin{pmatrix} \pm 1 - \frac{iyp}{\omega_0} \\ p \pm \frac{iy}{\omega_0} \\ -\frac{1+p^2}{\omega_0} \end{pmatrix}, \quad \frac{1}{\omega_0} \begin{pmatrix} p \\ \mp 1 \\ iy \end{pmatrix}, \quad \frac{1}{\sqrt{2}\sqrt{1+p^2}} \begin{pmatrix} \pm 1 + \frac{iyp}{\omega_0} \\ p \mp \frac{iy}{\omega_0} \\ \frac{1+p^2}{\omega_0} \end{pmatrix} \right\}.
\tag{27}
$$

Using those expressions in (23)-(24) yields to the first order corrections:

$$
\Omega_{1A}^\pm = \left\{ \mp \frac{1}{2(1+y^2+p^2)}, \mp \frac{1}{1+y^2+p^2}, \mp \frac{1}{2(1+y^2+p^2)} \right\},
\tag{28}
$$

$$
\Omega_{1B}^\pm = \left\{ \mp \frac{y^2}{(1+y^2+p^2)(1+p^2)}, 0, \mp \frac{y^2}{(1+y^2+p^2)(1+p^2)} \right\}.
\tag{29}
$$

The correction $\Omega_{1A}$ for the middle wave band is known as the dispersion relation of Rossby wave. The Berry correction $\Omega_{1B}$ is zero in that case, which is consistent with (24), together with $\Omega_0 = 0$ for this wave band. As explained previously, the correction $\Omega_{1B}$ for the other bands depends on the phase choice made in (27) for $\underline{\chi}_0$.

**Note on the literature.** The term $\Omega_{1A}$ will play a crucial role in ray tracing equations, in the next section. In that context, it was computed by N. Perez and called gradient correction, see (B15) in [13]. Using the same gauge choice, Y. Onuki obtained the expression of $\Omega_1$ as a particular case of a more general computing including horizontal variations of the layer thickness (equation (3.9) in [23]), without splitting the expression into a Berry part and a gauge-independent part. Gallagher and coworkers also obtained similar expressions in a series of paper on the mathematics of equatorial waves [18], including more general configurations with a prescribed horizontal mean flow field [19, 20].

## 3  Wave packet dynamics in the semi-classical regime

This section builds upon previous works on the manifestation of the Berry curvature in ray tracing as reviewed by Niu's group for electronic waves [10], Perez *et al* for geophysical waves [13], see also the pioneering work of Littlejohn and Flynn [9]. The only novelty here is a physical interpretation of the formal change of variable appearing in the paper by Littlejohn and Flynn, which allows us to make connections with Onuki's recent work on wave transport in geophysics [23].

### 3.1  Wave packet center of mass and wave packet momentum

We define the wave packet's location $y_v$ and wavenumber $p_v$ as the corresponding operators averaged with the energy density $\underline{\Psi}^\dagger \cdot \underline{\Psi}$ as weight, since the *multicomponent wave* field $\underline{\Psi}$ is the physical field to be observed:

$$
y_v = \int \mathrm{d}y \, \underline{\Psi}^\dagger \cdot (y\underline{\Psi}), \quad p_v = \int \mathrm{d}y \, \underline{\Psi}^\dagger \cdot (-i\epsilon \partial_y \underline{\Psi}),
\tag{30}
$$

where the subscript $v$ stands for *vectorial*, and where $\underline{\Psi}$ is normalized according to (10). In the quantum mechanical context, $y_v$ and $p_v$ are just the expectation values of the position and momentum operators. Recall that the normalization constraint (10) is equivalent to the energy conservation for shallow water waves. This is why $y_v$ and $p_v$ are respectively interpreted in this context as an averaged energy-weighted position and momentum (wavenumber) for a given

wavepacket. The weight $|\underline{\Psi}|^2$ corresponds indeed to the sum of local kinetic and available potential energy, which, in dimensional units, is $0.5(Hu^2 + Hv^2 + g\eta^2)$.

It will be useful to introduce similar quantities defined formally from the scalar wavefield $\psi$, although their physical interpretation is less straightforward:

$$y_s = \int \mathrm{d}y \; \psi^*(y\psi), \quad p_s = \int \mathrm{d}y \; \psi^*(-i\epsilon \partial_y \psi), \tag{31}$$

where the subscript $s$ stands for *scalar*. The quantities $(y_s, p_s)$ in (31) are averaged position and momentum, just as $(y_v, p_v)$ in (30), albeit with a different weight. In the case of $(y_v, p_v)$, the weight was the local energy density, a physically meaningful quantity. This is not the case for $(y_s, p_s)$. In fact, we will see in subsection 3.3 that $(y_s, p_s)$ are not physical observables for the wave-packet since they depend on the gauge choice for the wave vector reconstruction operator $\hat{\underline{\chi}}$ that relates the scalar field $\psi$ to the vector field $\underline{\Psi}$ through (9).

The two different definitions (30) and (31) for an averaged position and momentum of the wave packet can be related together. As we shall see, they are not equivalent, due to the non-commutation between the multicomponent projection operator $\hat{\underline{\chi}}$ and the operators $y$ or $\partial_y$. Defining

$$\mathbf{z} = (y, p), \quad \mathbf{z}_v = (y_v, p_v), \quad \mathbf{z}_s = (y_s, p_s), \tag{32}$$

and using the commutation relations (A.12-A.13) established in Appendix A.3, together with the Poisson bracket definition in (20), we find

$$\mathbf{z}_v = \mathbf{z}_s + i\epsilon \int \mathrm{d}y \; \psi^* \hat{\underline{\chi}}^\dagger \cdot \widehat{\{\mathbf{z}, \underline{\chi}\}} \psi. \tag{33}$$

From (30) to (33), no approximation is involved, as no assumption on the form of $\underline{\Psi}$ or $\psi$ is needed.

## 3.2 WKB ansatz for the scalar wavefield

We now consider the traditional WKB ansatz for the scalar wave field $\psi$:

$$\psi(y, t) = a_0 e^{\frac{i}{\epsilon}(\phi_0 + \epsilon \phi_1)} + \mathcal{O}(\epsilon), \tag{34}$$

where $a_0(y, t)$, $\phi_0(y, t)$ and $\phi_1(y, t)$ are real fields of order $\sim 1$. The normalization condition (10) for $\psi$ leads to the constraint

$$\int \mathrm{d}y \; a_0^2 = 1. \tag{35}$$

The scalar wave field $\psi$ is related to the multicomponent wave field $\underline{\Psi}$ through (9). Using the order zero expansion of operators acting on the WKB ansatz (as detailed in Appendix A.5) leads to

$$\underline{\Psi}(y, t) = a_0 e^{\frac{i}{\epsilon}(\phi_0 + \epsilon \phi_1)} \underline{\chi}_0(y, p(y, t)) + \mathcal{O}(\epsilon), \tag{36}$$

$$p(y, t) = \partial_y \phi_0 + \epsilon \partial_y \phi_1. \tag{37}$$

Using this WKB ansatz, the operators in (33) can be replaced by the expression of their symbols at leading order, in order to obtain an approximation at order $\epsilon$:

$$\mathbf{z}_v = \mathbf{z}_s + i\epsilon \int \mathrm{d}y \; a_0^2(y) \underline{\chi}_0^\dagger \cdot \{\mathbf{z}, \underline{\chi}_0\}\big|_{\mathbf{z}=(y,p)} + \mathcal{O}(\epsilon^2), \tag{38}$$

where the symbol $\underline{\chi}_0$ and the Poisson bracket are evaluated at point $(y, p)$, with $p(y, t)$ given by (37).

## 3.3 Assumption of a localized wave packet

We now assume that the wave packet is localized at $y_v$ and extends over a scale $\Delta y \ll 1$, keeping $\Delta y \gg \epsilon$. Using (35), equation (38) is further simplified into

$$\mathbf{z}_v = \mathbf{z}_s + i\epsilon \underline{\chi}_0^\dagger \cdot \left\{ \mathbf{z}, \underline{\chi}_0 \right\} + \mathcal{O}(\epsilon \Delta y^2), \tag{39}$$

where the terms in the r.h.s. are evaluated at $\mathbf{z}_s$. The expression remains actually correct when evaluated at $\mathbf{z}_v$, but corrections are then of order $\mathcal{O}(\epsilon \Delta y)$. From now on, we remove those correction terms. It is worth insisting on the importance of (39) by writing independently the two components, and by expanding the Poisson bracket:

$$y_v = y_s + i\epsilon \underline{\chi}_0^\dagger \cdot \partial_p \underline{\chi}_0, \tag{40}$$

$$p_v = p_s - i\epsilon \underline{\chi}_0^\dagger \cdot \partial_y \underline{\chi}_0. \tag{41}$$

The correction terms involve the components of a vector called the *Berry connection*:

$$\mathbf{A}(\mathbf{z}_s) = i\underline{\chi}_0^\dagger \cdot \nabla_{\mathbf{z}} \underline{\chi}_0, \quad \text{with} \quad \nabla_{\mathbf{z}} = (\partial_y, \ \partial_p). \tag{42}$$

Here, the Berry connection is a measure of how the polarization vector $\underline{\chi}_0(y_s, p_s)$ varies in phase space $(y_s, p_s)$. This term is gauge dependent: it is not invariant for a change of phase choice in $\underline{\chi}_0$. Since $y_v$ and $p_v$ are quantities built from the initial multicomponent wave problem that does not depend on any phase choice, they are gauge invariant. Thus, the breaking of gauge invariance by the Berry connection implies that the coordinates $(y_s, p_s)$ based on the scalar wave equation are *not* gauge-invariant. Those observations will be confirmed by the ray tracing equations obtained in both choices of coordinates.

Equations (40)-(41) correspond to the change of variable proposed by Littlejohn and Flynn to obtain gauge-independent phase space coordinates. [9]. We showed here that this change of variable has a wave packet interpretation.[6]

## 3.4 Ray tracing equations

### 3.4.1 Gauge dependent, canonical Hamiltonian form

A time differentiation of (31), together with the Hermiticity of the operator $\hat{\Omega}$ and the commutation rule (A.12), leads to

$$\dot{y}_s = \int \mathrm{d}y \, \psi^* \widehat{\partial_p \Omega} \psi, \quad \dot{p}_s = -\int \mathrm{d}y \, \psi^* \widehat{\partial_y \Omega} \psi. \tag{43}$$

Using the WKB ansatz in the limit $\epsilon \to 0$ followed by the localized wave packet assumption leads to the canonical ray tracing equations, where the frequency $\Omega_0 + \epsilon \Omega_1$ plays the role of the Hamiltonian,

$$\dot{y}_s = +\partial_{p_s} \Omega_s, \tag{44}$$

$$\dot{p}_s = -\partial_{y_s} \Omega_s, \tag{45}$$

$$\Omega_s(y_s, p_s) = \Omega_0(y_s, p_s) + \epsilon \Omega_{1A}(y_s, p_s) + \epsilon \Omega_{1B}(y_s, p_s), \tag{46}$$

where $\Omega_s$ is the symbol of the scalar operator. An unpleasant situation occurs: the ray dynamics in phase space $(y_s, p_s)$ depends on the term $\Omega_{1B}$ which is gauge-dependent, as $\Omega_{1B}$ depends on the phase choice for $\underline{\chi}_0$.

---

[6]The change of variable from scalar wave packet quantities $(y_s, p_s)$ to vectorial wave packet quantities $(y_v, p_v)$ correponds to the change of variables from $(\mathbf{x}, \mathbf{k})$ to $(\mathbf{x}', \mathbf{k}')$ with Littlejohn and Flynn's notations.

### 3.4.2 Gauge independent, non-canonical Hamiltonian form

Using the change of variables (40)-(41) in ray tracing equations (44)-(45)-(46) leads to a new set of equations:

$$\dot{y}_v = +\partial_{p_v}\Omega_v + \epsilon F_{yp}\dot{y}_v \,, \tag{47}$$

$$\dot{p}_v = -\partial_{y_v}\Omega_v + \epsilon F_{yp}\dot{p}_v \,, \tag{48}$$

$$\Omega_v(y_v, p_v) = \Omega_0(y_v, p_v) + \epsilon\Omega_{1A}(y_v, p_v) \,, \tag{49}$$

where $F_{yp}(y_v, p_v) = -F_{py}(y_v, p_v)$ is called the *Berry curvature*:

$$F_{yp} = i\{\underline{\chi}_0^\dagger, \underline{\chi}_0\} \,. \tag{50}$$

The Berry curvature is the curl of the Berry connection $\mathbf{A}(y, p) = (A_y, A_p)$ introduced in (42):

$$F_{yp} = \partial_y A_p - \partial_p A_y \,. \tag{51}$$

By contrast with the Berry connection, the Berry curvature is *gauge-independent*, as it is left unchanged by a change of phase choice for $\underline{\chi}_0$. In fact, all terms in the ray tracing equations (47)-(48)-(49) are gauge-independent. In particular, the term $\Omega_{1B}$ present in (46) has been cancelled out in (49) by a contribution induced by the change of variable from $(y_s, p_s)$ to $(y_v, p_v)$. This was expected as we already noticed that $y_v$ and $p_v$ are physical observable interpreted as averaged position and momentum with a local energy density weight, and the temporal evolution of such physical observables cannot be gauge-dependent. The price to pay for being gauge-independent in this new set of coordinates is that the presence of Berry curvature renders the Hamiltonian dynamics non-canonical [9]. Indeed, the system of equations (47)-(48) is Hamiltonian as it can be written at order $\epsilon$ in the form:

$$\dot{z}_i = J_{ij}\partial_{z_j}\Omega_v, \quad \text{with} \quad z = \begin{pmatrix} y_v \\ p_v \end{pmatrix}, \tag{52}$$

and where $J$ is the antisymmetric matrix

$$J = \begin{pmatrix} 0 & \frac{1}{1-\epsilon F_{yp}} \\ -\frac{1}{1-\epsilon F_{yp}} & 0 \end{pmatrix} \,. \tag{53}$$

It is not a canonical Hamiltonian system as $y_v$ and $p_v$ are not conjugate variables. The conserved "Hamiltonian energy" in phase space $y_v, p_v$ is the frequency $\Omega_v$. This conserved quantity is indeed left invariant by the change of phase space coordinates: $\Omega_v(y_v, p_v) = \Omega_s(y_s, p_s)$, up to order $\epsilon$, as noted by Littlejohn and Flynn [9].

The same set of gauge-invariant ray tracing equations can actually be obtained with a different derivation. Starting from a variational formulation of the linearized dynamics with a similar scaling, Perez *et al* recovered (47)-(49), and discussed geophysical applications. Our contribution here is to clarify the reason why the term $\Omega_{1B}$ that contributes to the symbol of $\hat{\Omega}$ does not enter into the ray tracing equations with gauge-independent phase space coordinates. Note that $\Omega_v$ is denoted by $\Omega$ in Perez *et al* [13].

*We conclude that the ray tracing equations describing the trajectories of the center of mass of the multicomponent wave involve only gauge-independent terms. The presence of Berry curvature corrections to those trajectories makes the Hamiltonian dynamics non-canonical.*

### 3.4.3 Application to shallow water waves

For shallow water waves described by the operators $\hat{\underline{\underline{\mathcal{H}}}}^{\pm}$, the Berry curvature is computed by using the expression (27) for $\underline{\chi}_0^{\pm}$ in (50). This yields an explicit expression of the Berry curvature for each of the three wave bands

$$F_{yp}^{\pm} = \left\{ \pm \frac{1}{\sqrt{1+y^2+p^2}^3}, \quad 0, \quad \mp \frac{1}{\sqrt{1+y^2+p^2}^3} \right\} . \tag{54}$$

As noticed in [13], there is no Berry curvature correction for the flat geostrophic wave band, but inertia-gravity wave packet trajectories are influenced by those corrections. We show in the next section that this Berry curvature is related to peculiar topological properties of the symbol of the rotating shallow water wave operator.

## 4 Topological properties of Matsuno symbol

The aim of this section is to interpret the Berry curvature (54) as a limiting case of a more general computation performed in [5], that highlights the topological origin of this quantity.

For that purpose, we need to step back to the original Matsuno shallow wave operator $\hat{\underline{\underline{\mathcal{H}}}}_k$ defined in (3), and to compute its symbol. This is done by noting that each term of the matrix operator in (3) is an elementary operator $\hat{g}(\tilde{y}, \partial_{\tilde{y}})$ whose symbol has already been computed in (A.4):

$$\underline{\underline{\mathcal{H}}}_k = \begin{pmatrix} 0 & i\tilde{y} & k \\ -i\tilde{y} & 0 & l \\ k & l & 0 \end{pmatrix}, \quad l = \frac{\tilde{p}}{\epsilon} . \tag{55}$$

Note that we have introduced the wavenumber $l = \tilde{p}/\epsilon$ in zonal direction, that will be more convenient to manipulate than the rescaled wavenumber $\tilde{p}$. Recall also that $\tilde{y} = |k|y$. One can then check that the symbols of the original shallow water wave operator and of the rescaled shallow water operators used previously in the semi-classical computations are related by $\underline{\underline{\mathcal{H}}}_k = |k|\underline{\underline{\mathcal{H}}}_0^{\pm}$. Note that there is no higher order corrections to the symbol, due to its simple form that involves no products of derivatives by functions of $y$. Such higher order corrections would emerge for instance when considering the effect of a varying bottom topography in this model, as detailed in [23, 24].

Our strategy now is (i) to compute the eigenvectors of this symbol matrix, (ii) to introduce the Berry curvature describing how the polarization relations of those eigenvectors change when parameters $(k, l, \tilde{y})$ are varied, (iii) to show that such Berry curvature is generated by a topological singularity in parameter space (iv) to recover the Berry curvature terms (54) as a limiting case. Those steps will be essential to bridge the gap between spectral flow properties of the operator $\hat{\underline{\underline{\mathcal{H}}}}_k$, and ray tracing equations deduced from operators $\hat{\underline{\underline{\mathcal{H}}}}^{\pm}$ in the semi-classical limit $k \to \pm\infty$.

### 4.1 Matsuno symbol and Kelvin plane waves.

The symbol matrix (55) can be interpreted as the Fourier representation of the wave operator considered in the original work of Kelvin for shallow water waves on the unbounded tangent plane to the sphere, assuming $\tilde{y} = f$ as a constant [31], see Fig. 3.

Delplace *et al* showed that the spectral flow associated with Matsuno wave operator for shallow water waves on the beta plane is encoded in the topological properties of eigenvectors

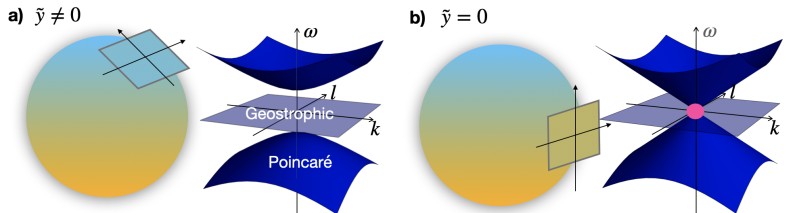

Figure 3: Dispersion relation of shallow water waves on a plane tangent to the planet, assuming constant Coriolis parameter $f = \tilde{y}$ (Kelvin wave problem). Geostrophic modes correspond to the Rossby modes of the Matsuno wave problem. The pink dot corresponds to the band degeneracy point that plays a central role in the topology of shallow water waves [5].

of those shallow water plane waves considered originally by Kelvin [5]. The Kelvin wave problem is simpler to solve than the Matsuno wave problem as it only involves the diagonalization of the $3 \times 3$ matrix (55).

**Existence of a degeneracy point.** We now assume that $\tilde{y}$ is a constant parameter. Except in the particular case $(k, l, \tilde{y}) = (0, 0, 0)$ The symbol matrix (55) admits three eigenvalues, namely 0 and $\pm\sqrt{k^2 + l^2 + \tilde{y}^2}$. These three eigenvalues define three distinct wave bands when $(k, l, \tilde{y})$ are varied, except at the particular point $(k, l, \tilde{y}) = (0, 0, 0)$. This point is peculiar as it corresponds to a degeneracy point for the eigenvalues, where the three wave bands touch each other. This band-touching point will play an important role later on.

### 4.2 Berry curvature for eigenvectors of Matsuno symbol

**Berry curvature in $(k, l, \tilde{y})$-parameter space.** An explicit expression for the normalized eigenvectors $\underline{\chi}(k, l, \tilde{y})$ of the symbol matrix (55) is given in [13]. From the knowledge of these eigenvectors, one can compute the Berry curvature **F**, which is conveniently expressed as a vector in parameter space $(k, l, \tilde{y})$:

$$\mathbf{F} = (F_{l\tilde{y}}, F_{\tilde{y}k}, F_{kl}), \tag{56}$$

where each component is obtained by applying the formula

$$F_{\lambda\mu} = i \frac{\partial}{\partial\lambda}\underline{\chi}^\dagger \cdot \frac{\partial}{\partial\mu}\underline{\chi} - i \frac{\partial}{\partial\mu}\underline{\chi}^\dagger \cdot \frac{\partial}{\partial\lambda}\underline{\chi}. \tag{57}$$

The Berry curvature of the three shallow water wave bands was computed in [5]:

$$\mathbf{F} = \left\{ -\frac{\mathbf{r}}{r^3}, \ 0, \ \frac{\mathbf{r}}{r^3} \right\}, \tag{58}$$

with $\mathbf{r} = (k, l, \tilde{y})$ and $r$ its norm. Those vector fields are displayed in Figs. 4a and 4b for negative and positive Poincaré wavebands, respectively.

Let us now explain how this Berry curvature vector is related to the Berry curvature introduced previously in (50).

**From $(k, l, \tilde{y})$-parameter space to phase space $(y, p)$.** The eigenvectors $\underline{\chi}_0^\pm(y, p)$ of $\underline{\underline{\mathcal{H}}}_0^\pm$ are given by the eigenvectors $\underline{\chi}(k, l, \tilde{y})$ of the matrix $\underline{\underline{\mathcal{H}}}_k$ defined in (55), using the change of variable

$$(k, l, \tilde{y}) = |k| (\pm 1, p, y). \tag{59}$$

The Berry curvature introduced in (50) for ray tracing equation is then recovered from the first component $F_{l\tilde{y}} = -F_{\tilde{y}l}$ of the Berry curvature introduced through (57), by expressing the semi-classical limit $\epsilon \to 0$ as a limiting case for $k$:

$$F_{yp}^{\pm} \, \mathrm{d}y \, \mathrm{d}p = \left[ \lim_{k \to \pm\infty} F_{\tilde{y}l} \right] \mathrm{d}\tilde{y} \, \mathrm{d}l \,. \tag{60}$$

This expression will play a crucial role in the next section, to relate the spectral properties found by quantization of ray trajectories to the topological properties in parameter space.

### 4.3 From Berry curvature to the first Chern number and Berry monopoles

**The first Chern number.** The Berry curvature introduced previously is a geometrical quantity, as it describes local properties of eigenvectors in parameter space. In loose terms, it describes how twisted is the eigenvector field locally, independently from any phase choice of the eigenvectors. Let us now introduce a topological invariant, the first Chern number, an integer that counts singularities in families of eigenvectors parameterized over a closed surface. Although more detailed and formal definitions of this invariant exist, we give below a definition of this number as an integral quantity involving the flux of Berry curvature across a closed surface.

**Chern-Gauss-Bonnet formula.** When considering eigenvectors parameterized over a 2D closed surface embedded in the 3D parameter space, one can compute a Berry flux induced by this curvature across any surface elements $\mathrm{d}a$ as $\mathbf{F} \cdot \hat{\mathbf{n}} \mathrm{d}a$, with $\mathbf{n}$ a unit vector normal to the surface, as displayed in Fig. 4. When integrated over the whole surface and normalized by $2\pi$, we get an integer, which is the first Chern number:

$$\mathcal{C} \equiv \frac{1}{2\pi} \int_{\mathcal{S}} \mathrm{d}a \, \mathbf{F} \cdot \hat{\mathbf{n}}, \quad \mathcal{C} \in \mathbb{Z} \,. \tag{61}$$

(61) is a generalization of the more familiar Gauss-Bonnet formula that relates the integral of the (geometrical) Gaussian curvature over a closed surface to the (topological) gender of this surface (the number of holes). Here, we consider a simply connected surface that can be continuously deformed to a sphere, and the Chern number appearing in (61) counts the number of phase singularities associated with the bundle of eigenvectors that are parameterized over the surface. This first Chern number is a topological index as it can not be changed under continuous deformation of the surface $\mathcal{S}$, provided that no degeneracy point is crossed during the deformation, just as the number of holes of a given closed 2D surface is not changed by continuously deforming this surface. Since the first Chern number is a topological invariant, singularities can be moved on a given surface (for instance by changing the phase choice for the eigenvectors), but can not be removed.

**Application to shallow water waves.** Coming back to the particular case of the shallow water wave problem, it turns out that only two configurations are possible depending on the surface $\mathcal{S}$, when computing (61) using the Berry curvature definition in Eqs (56)-(57): If the surface $\mathcal{S}$ encloses a degeneracy point, the Chern number can be nonzero. Otherwise, the Chern number is always zero. For this reason, it is sometimes said that the band degeneracy point carries a topological charge for a given wave band, whose value is given by the Chern number that can be computed either from (61), as done originally in [5], or by other methods [7]. Considering the parameter space $(k, l, \tilde{y})$ the result of the computation[7] is a triplet of Chern numbers associated with the three wave bands (by increasing order of frequency):

$$\mathcal{C} = \{-2, \ 0, \ 2\} \,. \tag{62}$$

---

[7] Note that the parameter space $(k, l, \tilde{y})$ is considered in ref. [7], hence the sign difference in the Chern numbers.

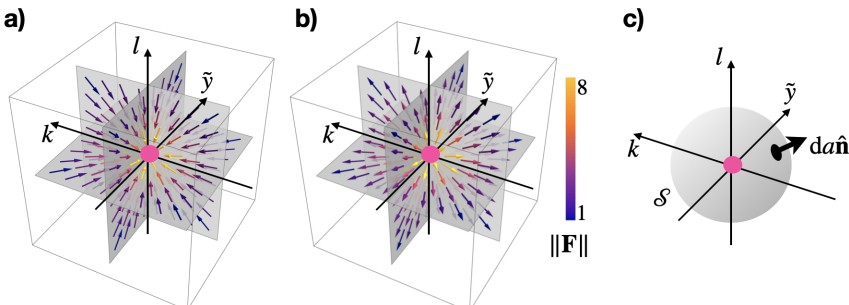

Figure 4: **a)** Berry curvature **F** vector field of the Matsuno symbol (55) for the negative Poincaré band. **b)** The same for the positive Poincaré band. Both diverge in amplitude at $(k, l, \tilde{y}) = (0, 0, 0)$. **c)** An integral of the flux of this vector field on any closed surface $\mathcal{S}$ enclosing this point evaluates to $-4\pi$ and $4\pi$, for the negative and positive Poincaré bands respectively. The pink dot represents a Berry-Chern monopole of charge $\pm 2$ that can be interpreted as the source of observed Berry curvature.

Thus, the positive-frequency Poincaré wave band is described by the first Chern number $\mathcal{C} = 2$. Our aim in the next section will be to explain how this topological invariant is related to the observed spectral flow of 2 in Matsuno spectrum through ray tracing dynamics in the semi-classical limit.

**Physical interpretation: analogy with a magnetic monopole.** As noted previously, the Berry curvature does not depend on the phase choice for the normalized eigenvector $\underline{\chi}$. It describes how fast the polarization relation changes when parameters are varied in the vicinity of a given point in parameter space. A direct consequence of (61) together with the existence of a degeneracy point carrying a non-zero Chern number is that the Berry curvature diverges close to the band-degeneracy points. In fact, one can interpret this curvature **F** as being generated by a *Berry monopole*, in the same way as a divergent or convergent magnetic field would be generated by a positive or negative magnetic monopole, whose charge must be quantized, according to a celebrated work by Dirac [6, 8]. In this analogy the first Chern number plays the role of the magnetic charge of the monopole. The Berry monopole can be interpreted as a hedgehogs topological point defect for the Berry curvature vector field **F**. This whole section on Berry curvature and Berry monopoles can now be summarized as follows:

*To conclude, the presence of a non-zero Berry curvature term in ray tracing equations is related to the presence of a Berry monopole at the origin of parameter space $(k, l, \tilde{y})$, where the three bands touch each others. This Berry monopole generates the Berry curvature away from the degeneracy point, and, in particular, generates the component $F_{yp}$ involved in (47)-(48) for ray trajectories in phase space $(y, p)$.*

## 5 From ray tracing to the spectrum of the wave operator

### 5.1 Bohr-Sommerfeld Quantization

Ray tracing can be used to find the set of eigenfunctions of the wave equation in the limit $\epsilon \to 0$. The standard semi-classical procedure is to look for closed orbits indexed by the frequency $\omega$ in phase space, and to select those orbits such that the phase gained by the wave

after a period (in phase space) is a multiple of $2\pi$, taking into account additional phase jumps of $\pm\pi/2$ picked up at turning points [15, 32]. Turning points occur where the wavenumber vanishes ($p = 0$). WKB expansion fails at such point, but the solution can be patched with another ansatz, and matching those two solutions in the asymptotic limit $\epsilon \to 0$ yields to a phase jump $\pi/2$ (see [32, 33]). Those phase jumps are related to a Maslov index [34]. To describe eigenmodes within the ray tracing framework, the phase $\phi$ originally defined in (34) is now assumed to be separated like $\phi(y, t) = \phi'(y) - \omega t$. In the following discussion we describe $\phi'(y)$ while dropping the prime to simplify notations.

In phase space with canonical coordinates $(y_s, p_s)$, ray trajectories with frequency $\omega$ are found by solving

$$\omega = \Omega_s(y_s, p_s = \partial_{y_s}\phi). \tag{63}$$

This is the equivalent of the Hamilton-Jacobi equation in classical mechanics, with the phase $\phi$ playing the role of an action. The phase picked up along a segment $dy$ is

$$d\phi = p_s(y_s)dy_s. \tag{64}$$

We explained in the previous section that computing trajectories in phase space $(y_s, p_s)$ is not easy and awkward, as the computation involves gauge-dependent terms. Following Littlejohn and Flynn [9], it is more convenient to consider phase space variable involving gauge-invariant coordinates $(y_v, p_v)$, with

$$\omega = \Omega_v(y_v, p_v), \tag{65}$$

where $\Omega_v$ is defined in (49). Using the relations (40)-(41), the integral transforms into

$$d\phi = p_v(y_v)\mathrm{d}y_v + i\epsilon \underline{\chi}_0^\dagger \cdot d\underline{\chi}_0 + \epsilon dg, \tag{66}$$

where $dg$ is the differential of the function $g = -p_v i \underline{\chi}_0^\dagger \cdot \partial_{p_v}\underline{\chi}_0$ in phase space. A proof of this can be found in [9] (p. 5249).

Let us now consider the case of a closed ray trajectory of frequency $\omega$, with two turning points. This will be the case of all ray trajectories for equatorial beta plane shallow water waves to be considered later. Let us compute the total phase $\Delta\phi$ picked up along a closed trajectory winding the origin once clockwise. First, we note that the term involving $dg$ vanishes. Second, we add the contributions from the WKB solutions to the phase jump picked up at turning points:

$$\Delta\phi = \oint_\omega p_v(y)dy + i\oint_\omega \epsilon \underline{\chi}_0^\dagger \cdot d\underline{\chi}_0 + \epsilon\pi, \tag{67}$$

where the contour integral are taken along the trajectory given by (65). For more complex trajectories with $\mu \in \mathbb{N}$ turning points, the additional term $\epsilon\pi$ should be replaced by $\epsilon\mu\pi/2$. The second term of the r.h.s. involves the Berry connection defined in (42). Using the Stokes theorem, it can be expressed as a flux of Berry curvature across the surface delimited by the closed trajectory at frequency $\omega$:

$$\Gamma(\omega) = \iint_\omega \mathrm{d}y_v \mathrm{d}p_v \, F_{py}(y_v, p_v), \tag{68}$$

where the integral is performed in the region delimited by the closed trajectory solution of (65).

Finally, the quantization condition is obtained by stating that the phase $\phi/\epsilon$ picked up along the trajectory must be an integer multiple of $2\pi$ to ensure that the WKB ansatz is single-valued. The quantization condition is finally expressed as

$$\oint_\omega p_v(y_v)\mathrm{d}y_v = 2\pi\epsilon\left(m + \frac{1}{2}\right) - \epsilon\Gamma(\omega), \quad m \in \mathbb{Z}. \tag{69}$$

Note that some values of $m$ may not be associated with solution of this equation. In fact, we show in the next section that $m$ admits a lower bound for shallow water waves.

**Note on the literature.** This quantization relation has long been used in condensed matter context to address the role of Berry curvature generated by a topological charge in shaping electronic waves, see e.g. [11]. Yet there seems to be so far little discussion on how to use those relation to describe the spectral flow. This is the aim of the next subsection, in the case of shallow water wave.

## 5.2 Application to shallow water waves

Let us now consider the two scalar operators $\hat{\Omega}^{\pm}$ describing the positive-frequency Poincaré wave band of $\underline{\hat{\mathcal{H}}}^{\pm}$ in the semi-classical limit $\epsilon \to 0$. To find the eigenvalues $\omega^{\pm}$ of the operators $\hat{\Omega}^{\pm}$, we apply the quantization procedure introduced in subsection 5.1.

Ray trajectories in phase space $(y_{\nu}, p_{\nu})$ are found by solving (65), using $\Omega_{\nu} = \Omega_0 + \epsilon \Omega_{1A}^{\pm}$, where $\Omega_0$ is given by (26) and $\Omega_{1A}^{\pm}$ is given by (28). This computation leads to circular trajectories of radius $\varrho(\omega)$, and eigenvalues $\omega^{\pm}$ are solutions of

$$\omega = \sqrt{1 + \varrho^2} \mp \frac{\epsilon}{2} \frac{1}{1 + \varrho^2}, \quad \varrho(\omega) = \sqrt{y_{\nu}^2 + p_{\nu}^2}. \tag{70}$$

Admissible values of $\omega^{\pm}$ are then obtained by applying the quantization relation (69), which leads to

$$m^{\pm} = \frac{1}{2\pi\epsilon} \oint_{\omega} p_{\nu}(y_{\nu}) \mathrm{d}y_{\nu} + \frac{1}{2\pi} \Gamma^{\pm}(\omega) - \frac{1}{2}, \quad m^{\pm} \in \mathbb{Z}, \tag{71}$$

where the integral in the r.h.s. of (71) is just the area inside the (clockwise) trajectory, and where the Berry flux is obtained by injecting in (68) the expression of the Berry curvature $F_{py} = -F_{yp}$ given in (54):

$$\oint_{\omega} p_{\nu}(y_{\nu}) \mathrm{d}y_{\nu} = \pi \varrho^2(\omega), \quad \frac{1}{2\pi} \Gamma^{\pm} = \pm \left( 1 - \frac{1}{\sqrt{1 + \varrho^2(\omega)}} \right). \tag{72}$$

The functions $\varrho(\omega)$ and $\Gamma^{\pm}(\omega)$ for the positive frequency Poincaré wave band are displayed in Fig. 5a and 5b, respectively. Our aim is now to compare the spectra of $\hat{\Omega}^+$ and $\hat{\Omega}^-$, taking advantage of the quantization condition in (71).

We need to choose a convention to index the eigenfunctions of both operators, in such a way that it may be possible to pair them together based on some physical criterion. For that purpose, it is useful to consider the limit of large frequency $\omega \to +\infty$. In this limit, the Berry curvature terms $\Gamma^{\pm}$ tend to $\pm 2\pi$, and the phase space trajectories $\omega = \Omega_{\nu}^{\pm}(y_{\nu}, p_{\nu})$ become identical at order $\epsilon$. Indeed, the symbols $\Omega^{\pm}$ are expressed as $\omega_0 + \epsilon \Omega_1^{\pm} + \mathcal{O}(\epsilon^2)$, with $\Omega_1^{\pm} = \Omega_{1A}^{\pm} + \Omega_{1B}^{\pm}$ given in (28)-(29). A direct inspection of those first-order correction terms for the positive-frequency Poincaré wave band shows that they vanish in the large frequency limit, for which $\varrho(\omega) \to +\infty$. This implies

$$\lim_{\omega \to +\infty} \left( m^+ - m^- \right) = 2. \tag{73}$$

Because of the convergence of their symbol to a common expression, spectral properties of the operators $\hat{\Omega}^{\pm}$ are also expected to converge in the large frequency limit. The pairing procedure between eigenmodes of $\hat{\Omega}^+$ and $\hat{\Omega}^-$ can thus formally be performed by assigning a common index $n$ to those modes in the large frequency limit. This is done by choosing

$$m^{\pm} = n \pm 1. \tag{74}$$

The term $\pm 1$ cancels the Berry curvature terms in (71) in the large frequency limit.

While the WKB solution is valid in the limit $\epsilon \to 0$ for a given value of $\varrho$, the quantization condition (71) is not guaranteed in the limit $\varrho \to 0$, for a given $\epsilon$, as one can not distinguish anymore phase jumps at the turning points from the phase gained by the WKB solution. More precisely, (71) is valid with the scaling $\varrho \gg \sqrt{\epsilon}$, that makes possible a scale separation between the WKB part of the solution and the phase picked up at the turning points. Fortunately, in the limit $\varrho \to 0$ the operators $\hat{\Omega}^\pm$ are described by shifted quantum harmonic oscillator operators, which allows us to find an explicit computation for their eigenmodes at finite $n^\pm$, as shown in Appendix B. This part of the spectrum can then be matched to the semi-classical computation presented in this section, as the quantum harmonic oscillator soution remains valid for $\varrho \sim \epsilon^\alpha$ with $0 < \alpha < 0.5$. An important outcome of the quantum harmonic oscillator computation is the condition:

$$n \geq 1, \quad \text{for eigenfunctions of } \hat{\Omega}^-, \text{ ensuring } m^- \in \mathbb{N}, \tag{75}$$

$$n \geq -1, \quad \text{for eigenfunctions of } \hat{\Omega}^+, \text{ ensuring } m^+ \in \mathbb{N}. \tag{76}$$

In fact, these results could have directly been deduced from Bohr-Sommerfeld rule (71) by noting that the area term must be positive and that $\Gamma^\pm = 0$ in the limit $\varrho \to 0$, as displayed Fig. 5. The constraints (75)-(76) mean that two modes labelled by $n = -1$ and $n = 0$ are unpaired! This was precisely expected from the bulk-boundary correspondence [5,7]. In fact, one can check that

- $n = -1$ is the Kelvin mode,
- $n = 0$ is the Yanai (or mixed Rossby-gravity) mode,

and that $n \geq 0$ counts the number of zeros for the field $v$, which is an invariant of a given branch in the dispersion relation of equatorial shallow water waves [1].

*To conclude, ray tracing and quantization condition in the limit $\epsilon \to 0$ allow us to recover a semi-classical version of the spectral flow result: just as two modes are gained by the positive-frequency Poincaré wave band as $k$ is varied from $-\infty$ to $+\infty$, the operator $\hat{\Omega}^+$ admits two more eigenmodes than the operator $\hat{\Omega}^-$ in the semi-classical limit $\epsilon \to 0$.*

## 5.3 Mode imbalance interpreted by Chern-Gauss-Bonnet formula

The analysis of subsection 5.2 shows that the imbalance of 2 modes between operators $\hat{\Omega}^+$ and $\hat{\Omega}^-$ in the semi-classical limit is due to the Berry curvature flux corrections $\Gamma^\pm$ involved in the Bohr-Sommerfeld quantization conditions, with

$$\lim_{\omega \to +\infty} \frac{\Gamma^+(\omega) - \Gamma^-(\omega)}{2\pi} = 2, \tag{77}$$

as illustrated graphically in Fig. 5b. We explained in subsection 5.2 that the Berry curvature term $F_{yp}^\pm$ involved in the expression of $\Gamma^\pm$ in (68) is related to the presence of a Berry monopole in parameter space $(k, l, \tilde{y})$ for the eigenvectors of the symbol matrix (55).

In fact, (77) can be interpreted as the direct outcome of the Chern-Gauss-Bonnet formula (61). To show this, let us consider the parameter space $(k, l, \tilde{y})$, and the closed cylindrical surface $\mathcal{S}$ depicted Fig. 6. The length of the cylinder is $2k = 2/\sqrt{\epsilon}$, while its circular ends are delimited by a closed circular ray trajectory with a radius $\varrho(\omega)$ in phase space $(\pm 1, y, p)$, which is related to parameters $(k, l, \tilde{y})$ through the relation (59), where the index $\nu$ for phase space variables $(y, p)$ has been dropped for convenience.

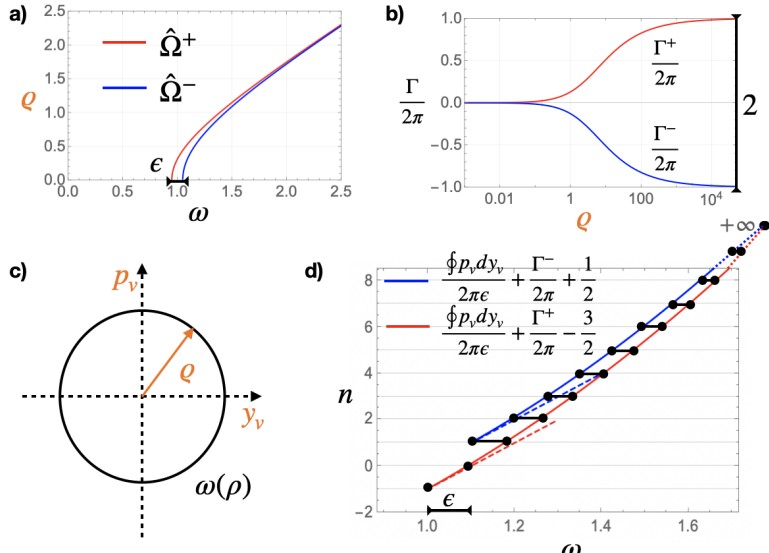

Figure 5: **(a)** Variations of the circular trajectory radius $\varrho$ as a function of frequency $\omega$ in phase space $(y_v, p_v)$. **(b)** Variation of the Berry flux across the area delimited by a circular ray trajectory having a radius $\varrho$. **(c)** A circular trajectory in phase space. **(d)** Quantization condition for shallow water waves in the semi-classical limit, as expressed in (71)-(74). Red and blue curves are associated respectively with operators $\hat{\Omega}^+$ and $\hat{\Omega}^-$. The sign $\pm$ is due to the definition $n^\pm = 1$ Dashed lines correspond to explicit computations performed in the limit $\epsilon \to 0$ at finite $n$, see Appendix B.

The surface $\mathcal{S}$ encloses the degeneracy point $(0, 0, 0)$. Thus, according to Chern-Gauss-Bonnet formula (61), the Berry flux across this surface normalized by $2\pi$ is equal to the Chern number carried by this degeneracy point for each wave bands, which are given by (62) for the shallow water case. The normalized Berry flux can be decomposed into three contributions:

$$\frac{1}{2\pi} \int_{\mathcal{S}} da \mathbf{F} \cdot \mathbf{n} = \frac{1}{2\pi} \int_{\mathcal{S}_{cyl}} da \mathbf{F} \cdot \mathbf{n} + \frac{1}{2\pi} \int_{\sqrt{y^2+p^2} \le \varrho} dy \, dp \, F_{py}^+ - \frac{1}{2\pi} \int_{\sqrt{y^2+p^2} \le \varrho} dy \, dp \, F_{py}^-, \quad (78)$$

where $\mathcal{S}_{cyl}$ represents the open cylindrical surface, and where we have used $F_{\tilde{y}l} df \, dl = F_{yp} dy \, dp$, consistently with the definition of Berry curvature in (57).

To estimate those three contributions to the Berry curvature flux, we consider a double limit whose order matters: first, the limit of infinite radius $\varrho \to +\infty$ for a given value of $k$ second, the semi-classical limit $k \to \pm\infty$. The motivation for this double limit comes from the fact that we want to interpret how global properties of the full spectrum are changed when $k$ is varied. The first limit allows to get properties of the full spectrum for a given value of $k$, since this limit allows to scan all possible trajectories in phase space $(y, p)$. The second limit allows us to get asymptotic properties of this full spectrum in the semi-classical limit.

The first limit $\varrho \to +\infty$ implies that the contribution of Berry flux across the open cylinder surface $\mathcal{S}_{cyl}$ vanishes, as the Berry curvature vanishes at infinite distance from the origin in parameter space [5]. More precisely, the Berry curvature decreases as $\varrho^{-3/2}$, which is faster than the increase in the area $\mathcal{S}_{cyl} \sim \varrho k$. Once this limit has been taken, the only non-zero contribution to the Berry flux across $\mathcal{S}$ comes from the two circular surfaces at the ends of the cylinder. The Berry flux across those surfaces only involve the component $F_{yp}$ which tends to $F_{yp}^\pm$ computed in (54) when taking the limit $k \to \pm\infty$. We note that the limit $\rho \to +\infty$ is equivalent to the limit $\omega \to +\infty$ for positive-frequency Poincaré waves, and that $k \to \pm\infty$ is

equivalent to the semi-classical limit $\epsilon \to 0$. Finally, we get

$$\lim_{\epsilon \to 0} \lim_{\omega \to +\infty} \frac{1}{2\pi} \int_{\mathcal{S}} \mathrm{d}a \mathbf{F} \cdot \mathbf{n} = \lim_{\omega \to +\infty} \frac{\Gamma^+(\omega) - \Gamma^-(\omega)}{2\pi}, \tag{79}$$

which is equal to a first Chern number $\mathcal{C} = 2$, according to the Chern-Gauss-Bonnet formula (61) and expression (62) for the Chern number in shallow water case. This shows the topological origin of the integer number 2 in the r.h.s. of (77). A last remark is in order:

*Our point here is to stress that ray tracing followed by quantization in an appropriate semi-classical limit offers a physically appealing intuitive explanation on the relation between the topological index and the spectral properties of the operator. In that respect, it may complement previous lecture notes on this topic [7, 8]. While the derivation has been focused on the shallow water wave problem, the method illustrated with this example applies to a much broader class of wave problems.*

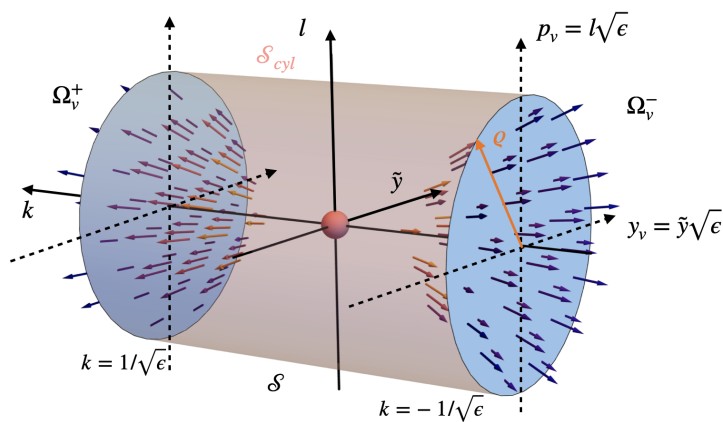

Figure 6: **Parameter space and phase space for ray tracing**. A Berry monopole is located at the origin of $(k, l, \tilde{y})$-parameter space, which corresponds to a degeneracy point for the eigenvalues of matrix $\overline{\overline{\mathcal{H}}}_k$. Phase space for ray tracing associated with the scalar wave operators $\hat{\Omega}^\pm$ are recovered in the semi-classical limit $\epsilon \to 0$, with the change of variable from $(l, \tilde{y})$ to $(p_v, y_v)$ given in (59). The surface $\mathcal{S}$ in parameter space is a cylinder where both circular ends are delimited by closed phase space trajectories of radius $\varrho$ at positions $k = \pm 1/\sqrt{\epsilon}$.

## 5.4 Spectral flow and phase space density of states

**Liouville theorem.** We noticed that ray tracing equations (47)-(48) in phase space $(y_v, p_v)$ are an example of non-canonical Hamiltonian systems, which are commonly encountered in fluid dynamics [35]. For such systems, the equivalent of standard Liouville theorem holds [35]. To see this, let us start with the canonical Hamiltonian system (44)-(45) for trajectories in phase space $(y_s, p_s)$. In that case phase space volume element $\delta \mathcal{V} = \mathrm{d}y_s \mathrm{d}p_s$ is trivially conserved owing to the canonical structure of the ray tracing equations. Now, the non-canonical ray tracing equations (47)-(48) were derived from this canonical set of equations by performing the change of variable (40)-(41). Applying this change of variable to the volume element then yields to the conservation of

$$\delta \mathcal{V} = \det \left( \frac{\partial(y_s, p_s)}{\partial(y_v, p_v)} \right) \mathrm{d}y_v \mathrm{d}p_v = (1 - \epsilon F_{yp}) \mathrm{d}y_v \mathrm{d}p_v. \tag{80}$$

This means that a Liouville theorem holds in phase space $(y_\nu, p_\nu)$, albeit with a modified density of states that depends on the Berry curvature.

**Counting the number of states.** Owing to the uncertainty principle, a given state of the system occupies at least the elementary phase space volume $2\pi\epsilon$. According to standard arguments, this elementary phase space volume just reflects the impossibility to have a perfectly localized wave packet with a given wavenumber [36]. One can then count the total number of states that can be hosted in phase space as the integral of $\delta\mathcal{V}$ over phase space divided by the elementary phase space volume $2\pi\epsilon$. Here, this number is infinite. However, just as one can compute mode imbalance between operators $\hat{\Omega}^+$ and $\hat{\Omega}^-$, we can compare the number of modes that are hosted in phase space $(y_\nu, p_\nu)$ for ray trajectories ruled by $\hat{\Omega}^+$ and $\hat{\Omega}^-$. The corresponding volume elements are denoted by $\delta\mathcal{V}^\pm$, and a direct application of (80) leads to:

$$\frac{1}{2\pi\epsilon}\left(\int \delta\mathcal{V}^+ - \delta\mathcal{V}^-\right) = \frac{1}{2\pi}\int \mathrm{d}y_\nu \mathrm{d}p_\nu \left(F_{yp}^- - F_{yp}^+\right) = 2, \tag{81}$$

where the integrals are performed over the whole phase space $(y_\nu, p_\nu)$. The last equality follows from previous computations involving (54).

We see that the mode imbalance between operators $\hat{\Omega}^+$ and $\hat{\Omega}^-$ can be interpreted as a consequence of the continuous deformation of the density of state $(1 - \epsilon F_{yp}^\pm)/(2\pi\epsilon)$ induced by the Berry curvature in phase space $(y_\nu, p_\nu)$. In condensed matter context, this phenomenon has been realized and popularized by Niu and coworkers [10]. In fact, the modification of phase space density has long been known by physicists working with non canonical hamiltonian systems, e.g. [35]. Yet the relation with a spectral flow was not explicit in those previous studies. This relation can now be stated as follows for the shallow wave problem:

*The Berry curvature induced by a Chern-Berry monopole of charge 2 in $(y, p, k)$-space modifies phase space density in $(y_c, p_c)$-space such that 2 more states (among an infinite number) are hosted in the limit $k \to +\infty$ than in the limit $k \to -\infty$.*

## 6 Discussion and conclusion

### 6.1 Main physical ideas underlying the bulk-interface correspondence

By considering the particular case of equatorial waves, we have proposed a novel physical interpretation of the interface-bulk correspondence that is commonly invoked to interpret the wave spectrum of continuous media with spatially varying coefficients. Let us summarize the main points of the argument to emphasize the physical concepts that have been introduced along the way and to show how they are related to previous work.

In the case of equatorial shallow water waves, Mastuno discovered in 1966 a spectral flow of index 2, corresponding to two modes gained by the upper frequency waveband when the zonal wavenumber $k$ was varied from $-\infty$ to $+\infty$ [1]. He found that those modes were more localized than the others along the equator, and that they only propagate energy eastward. In 2017, [5] noticed that this spectral flow index is related to a Chern number describing singularities in families of plane wave solutions for the same shallow water flow model albeit in a simpler $f$-plane configuration, where the Coriolis parameter is held constant in space, and thus considered as an external parameter. Concretely, [5] computed this Chern number carried by a three-fold degeneracy point for the $f$-plane wave bands in $(k, f, p)$ parameter space, with $p$ the wavenumber in meridional direction, and interpreted it as the charge of a Berry monopole. The authors concluded that global properties of the *dispersion relation* in the the beta-plane

(complicated) problem could be predicted just by computing the *polarization relation* of the (simpler) $f$-plane wave problem, as expected from an interface-bulk correspondence and index theorems [7].

Mastuno's computation was performed for arbitrary wavenumber $k$ with brute force analytical computation. Here, we have recovered Matsuno's result in the particular limit $|k| \rightarrow +\infty$. The interest of considering this particular limit is to take advantage of the small zonal wavelength parameter to perform a multiple scale development akin to semi-classical or WKB analysis. In this particular situation, the semi-classical computation is less elegant that the direct solution of the problem valid for all $k$, but is extremely useful to get a physical interpretation of the result, and to be applied to a much wider class of wave problems.

In the semi-classical limit, the computation of the wave spectrum boils down to computing trajectories of wave-packets in phase space $(y, p)$, and to select trajectories such that the phase picked up by a wave-packet along a closed contour is an integer multiple of $2\pi$.

At lowest order, local properties of the wave-packet trajectory at point $(y, p)$ in phase space are obtained by assuming a local plane-wave solution, with $f(y)$ considered constant at the scale of the wave-packet. This local plane wave solution comes along with a polarization relation, and a dispersion relation, which are obtained by solving a matrix problem, referred to as the bulk problem – or the symbol in mathematics. As explained by [7] for a more general class of wave problems exhibiting spectral flows, this procedure makes a direct connection between the parameter space $(k, f, p)$ considered in [5], and the phase space $(y, p)$ for wave-packets with zonal wavenumber $k$ sufficiently large. Wigner-Weyl transforms and symbolic calculus is just a standard technical way to formalize and perform this translation from the operator to the phase space with local plane wave solutions.

Our contribution here is to explicitly compute the ray trajectories in phase space for a wave-packet in a given wave band, including first order corrections within the semi-classical expansion framework. The key point of our analysis is thus to compute the wave trajectory taking into account a small correction in the polarization that results in asymmetry for phases of left-ward and right-ward propagating waves. The formal development leading to those first order corrections can be found in previous studies following the seminal work of Littlejohn and Flynn [9, 10, 12, 13].

Once trajectories are computed, the second key step is to use the Bohr-Sommerfeld selection rule to find the discrete set of wave-packet trajectories that correspond to eigenmodes of the equatorial beta-plane shallow water wave problem, and to compare the sets of eigenmodes in the limit $k \rightarrow +\infty$ and $k \rightarrow -\infty$, for the upper frequency Poincaré wave band. Any mismatch is interpreted as a spectral flow for varying $k$. Indeed, wave branches in the dispersion relation can neither be created nor annihilated as $k$ is varied [37]. The reason is that the wave operator is self-adjoint for any given $k$ and admits only discrete spectrum. Thus it is possible to follow the different dispersion branches by varying $k$. The set of eigenmodes for any value of $k$ defines a complete orthonormal basis for triplet of 1D fields in $y$ direction. Thus, if a wave-band has more modes for $k \rightarrow +\infty$ than for $k \rightarrow -\infty$, it means that the additional modes have transited from other wave-bands as $k$ was varied.

The phase picked up by the wave-packet along a closed trajectory in phase space $(y, p)$ involves two contributions, which can be interpreted as the dynamical and geometrical (Berry) phase.

The dynamical phase is just the phase that one would expect from inspection of the dispersion relation. This dispersion relation is at lowest order the dispersion of the $f$-plane problem with the value of $f$ at the averaged location of the wave packet. This dispersion relation is the

same for $k > 0$ and $k < 0$, and thus involve no asymmetry in the selection rule.

The geometrical phase corresponds to a mismatch of the (complex) polarization phase of the eigenvector after circulating along the closed orbit. This is very much like the angle gained by the pendulum in Foucault experiment after a day, after the system has completed a closed trajectory along a latitude circle. In Foucault pendulum case, the angle increases from 0 to $2\pi$ when varying the latitude from the 0 to the North pole. Similarly, for shallow water wave problem in phase space with $k > 0$, the geometrical phase picked up by the wave packets also increases from 0 to $2\pi$ when the radius of ray trajectory increases from 0 to infinity. The opposite phase is picked up for $k < 0$. The geometrical corrections in phase space $(y, p)$ are directly related to the Berry monopole described and computed in 2017 by [5] in parameter space $(k, f(y), p)$. The key message here is that the phase difference of $4\pi$ corresponds both to this Berry-Chern monopole and to the mismatch of two modes in the Poincaré wave band for $k \to +\infty$ and $k \to -\infty$, and ray tracing bridge the gap between those two point of views.

It should be stressed here that gradients of the Coriolis parameter $f$ involve additional corrections to the dynamical phase that we computed, building upon [9, 10, 13], and that are of the same order as the corrections due to the geometrical phase. It turns out that such corrections do not lead to a net gain or loss of modes when counting them at a given value of $k$, even if they play a prominent role in setting the value of the frequency levels. Such corrections may break the $k \leftrightarrow -k$ symmetry, but we showed that they actually vanish for closed orbits with a sufficiently large radius.

Another subtlety of the approach is that the Bohr-Sommerfeld rule applies to trajectory with sufficiently large radius. We explained that this regime could be matched asymptotically with another regime for which the wave operator admits explicit solutions, corresponding to the eigenmodes of a shifted scalar quantum harmonic oscillator. Again, the precise value of the frequency levels depends on this procedure, but not the number of modes.

The ray tracing argument thus explains how a topological defect in parameter space leads to a mismatch of wavemode numbers for the spectrum in the semi-classical limit. It also predicts that the modes involved in the spectral flow have a lower index than the others, meaning that the modes are more localized than the others at a given $k$, since the trajectories in phase space associated with those modes are closer to the origin. It would be interesting to show in this ray tracing framework that a deformation of the beta plane into a smooth step profile for $f(y)$ would indeed lead to the delocalization of all the modes but those two additional modes of topological origin, as described for instance in [38] by computing explicitly the spectrum in a solvable case.

We also showed along the way that the noncanonical structure of the Hamiltonian ray equations allows for a second, physically appealing interpretation of the spectral flow: in the limit of large wavenumber $|k|$, the density of state in phase space $(y, p)$ is modified in such a way that two more states can be accommodated for $k > 0$ than for $k < 0$. Again, this density of state is governed by the presence of a Berry monopole in parameter space.

In that respect, our study provides a physical interpretation of the index theorem, complementary to more rigorous approaches [7]. In fact, our study relates a spectral flow index to a topological index, while Atiyah-Singer theorem involves an analytical index, which, in the shallow water case and other relevant physical systems, has been suitably defined and interpreted in a recent study by Delplace [8]. In the present paper, a semi-classical approach was used to reconstruct formally spectral properties of a wave problem with a parameter varying in a given direction. On a more rigorous side, semi-classical technique has been recently used in mathematical context to describe how a given wave packet evolving between two topologically distinct materials splits into a "bulk" part that spreads and eventually collapses and "edge" part that remains coherent and propagates at the interface between the two topological phases [39].

## 6.2 Perspectives

We stress that we focused on a semi-classical interpretation of a *bulk-interface* correspondence, in problems without boundaries. In such problems, waves of topological origin emerge along an interface of a parameter that opens a frequency gap for the bulk waves, as the Coriolis parameter for equatorial waves. While the method was presented in the case of equatorial shallow water waves, it is sufficiently general to be applied to a much wider class of problems including more general shallow water waves [24,52], plasma [27,28], continuously stratified and compressible flows [25,26], among others. In all those problems with Hermitian wave operator, topological features are found by looking for Berry monopole in parameter space, and we have shown how such monopole may affect wave-packet dynamics in phase space. The *bulk-interface* correspondence discussed in this paper is different than the *bulk-boundary* correspondence that is commonly encountered in condensed matter. In the case of a bulk-boundary correspondence, a bulk Chern number can be defined on each side of the interface (e.g. each side of the equator for shallow water waves). Assigning a topological invariant on each side of the interface for continuous media is not always possible. For instance, for rotating shallow water waves, it requires the introduction of a regularization parameter such as odd-viscosity [38,41–43]. When this parameter is added into the model, one can assign a topological index to the $f$-plane problem, i.e. in each hemisphere, and predict the number of unidirectional modes at the equator [38]. The case of sharp (discontinuous) interfaces or hard boundaries such as solid walls is much more complicated for continuous media: unidirectional modes exist [43], yet apparent violation of standard bulk-boundary correspondence have been noted and explained [44,45]. Our present work applies to the unbounded case and as such provide an explanation for the bulk-interface correspondence only. However, hard-boundary cases may sometimes be interpreted as limiting cases of an interfaces. For instance, unidirectional trapped modes along coasts found by Kelvin in 1880 [31] can be recovered from the study of a shallow water with varying bottom topography defining an interface at the coast [24]. In that context, the ray tracing approach may help to bridge the gap between the modern topological point of view and the more traditional textbook skipping orbit picture explaining the emergence of the unidirectional mode along the wall. In that context of coastal waves, it will be interesting to relate the ray tracing approach to the interpretation of rotating shallow water dynamics as a Chern-Simons topological field theory [46].

It will also be interesting to see in future work how non-Hermitian wave problems can be tackled with the ray tracing approach. Such problems naturally occur in the context of flow instabilities [47], or in the presence of dissipative terms in the dynamics. On the one hand, some of the spectral flow properties found in the Hermitian case seem to be robust to the presence of non-Hermitian effects, as reported in the context of shallow water dynamics in the presence of a mean velocity shear [52]. On the other hand, intriguing new unidirectional trapped modes have been reported in rotating convection [48,53,54]. The ray tracing point of view may help to bring new hindsight on the topological origin of those phenomena.

## Acknowledgements

We warmly thank Pierre Delplace for many insightful discussions and suggestions on this topic. This work was supported in part by the Collaborative Research Program of Research Institute for Applied Mechanics, Kyushu University, and in part by the national grant ANR-18-CE30-0002-01. N.P. was funded by a PhD grant allocation Contrat doctoral Normalien.

# A  Symbolic calculus in a nutshell

This appendix gives the definition and some important properties of the Weyl-Wigner transform that are used for symbolic calculus. It follows closely the presentation given in Onuki 2020 [23], we chose to present them here to set our notations.

## A.1  Wigner-Weyl transform

Let us consider a linear operator $\hat{f}$ that can formally be written with an integral representation as

$$\hat{f}\psi(y) = \int \mathrm{d}y'\, F(y, y')\psi(y'). \tag{A.1}$$

The symbol of this operator is a phase-space function $f(y, p)$ obtained through the Wigner transform:

$$f(y, p) = \int \mathrm{d}y'\, F\left(y + \frac{y'}{2}, y - \frac{y'}{2}\right) e^{-\frac{i}{\epsilon}py'}. \tag{A.2}$$

Knowing the symbol $f(y, p)$, the operator $\hat{f}$ is recovered through the Weyl transform defined as

$$\hat{f}\psi(y) = \frac{1}{2\pi\epsilon} \int \mathrm{d}y'\mathrm{d}p\; e^{i\frac{p}{\epsilon}(y - y')} f\left(\frac{y + y'}{2}, p\right)\psi(y'). \tag{A.3}$$

This process is sometimes called Weyl quantization, as it is a way to deduce a quantum operator from a classical phase-space function. The origin of this quantization procedure is given in subsection A.4.

## A.2  Simple examples

Here, in geophysical fluid dynamics, our starting point is the operator of the linearized flow dynamics, writen formally as $\hat{f}(y, \partial_y)$. The integral Kernel of the operator $F(y, y')$ is in general not known *a priori*, and, in fact, it does not need to be known to compute the symbol $f(y, p)$, in most practical situation. Indeed, the use of equation (A.3), together with one or several integration by parts makes it possible to derive the following useful results:

$$\begin{aligned}
\text{Symbol} &\longleftrightarrow \text{Operator}, &\tag{A.4}\\
g(y, p) &\longleftrightarrow \hat{g}(y, \partial_y), &\tag{A.5}\\
y &\longleftrightarrow \hat{y} = y, &\tag{A.6}\\
p &\longleftrightarrow \hat{p} = -i\epsilon\partial_y, &\tag{A.7}\\
c(y)p &\longleftrightarrow \widehat{c(y)p} = -i\epsilon c(y)\partial_y - i\epsilon\frac{c'(y)}{2}. &\tag{A.8}
\end{aligned}$$

In the last line, $c(y)$ is a sufficiently smooth function and $c' = dc/dy$ is its derivative. This relation is important as it shows that products of symbols are in general different than products of operators. We come back to this important point in the next subsection.

Up to now we have only discussed the case of scalar operators and symbols. It is straightforward to extend the definition of Wigner-Weyl transforms to matrix symbols and corresponding operators, which are matrices where each component is a scalar operator. One then gets the

symbol of the shallow water model operator:

$$\text{Symbol} \quad \longleftrightarrow \quad \text{Operator} \tag{A.9}$$

$$\underline{\underline{\mathcal{H}}}(y,p) \quad \longleftrightarrow \quad \underline{\underline{\hat{\mathcal{H}}}}(y,\partial_y) \tag{A.10}$$

$$\begin{pmatrix} 0 & iy & 1 \\ -iy & 0 & p \\ 1 & p & 0 \end{pmatrix} \quad \longleftrightarrow \quad \begin{pmatrix} 0 & iy & 1, \\ -iy & 0 & -i\epsilon\frac{\partial}{\partial y}, \\ 1 & -i\epsilon\frac{\partial}{\partial y} & 0 \end{pmatrix}. \tag{A.11}$$

### A.3 Products and commutation rules

One can also deduce from the definition of Weyl transform and an integration by part that

$$y\hat{f} - \hat{f}y = i\epsilon\widehat{\partial_p f}, \tag{A.12}$$

$$\partial_y\hat{f} - \hat{f}\partial_y = \widehat{\partial_y f}. \tag{A.13}$$

In general, taking the Weyl transform of a symbol product $f g$ does not lead to to the standard product of corresponding operators $\hat{f}\hat{g}$. It is however possible to define a new product operator at the level of symbol, called star product, or Moyal product, such that

$$\hat{f}\hat{g} = \widehat{f \star g}. \tag{A.14}$$

It follows from the definition of the Wigner-Weyl transform, a Taylor expansion and some manipulations detailed in the next subsection that

$$f \star g = \sum_{(n,l)\in\mathbb{N}^2} \frac{(-1)^n}{n!l!} \left(\frac{i}{2}\epsilon\right)^{n+l} \left(\partial_y^l \partial_p^n f\right)\left(\partial_y^n \partial_p^l g\right). \tag{A.15}$$

Now, we assume $\epsilon \to 0$. Up to order one, the star product is

$$f \star g = f g + \frac{i}{2}\epsilon\{f,g\} + \mathcal{O}\left(\epsilon^2\right), \tag{A.16}$$

which is in practice the expression used in this paper for the star product.

A useful consequence is the commutation rule used in the main text:

$$\hat{f}\hat{g} - \hat{g}\hat{f} = i\epsilon\widehat{\{f,g\}} + \mathcal{O}\left(\epsilon^2\right). \tag{A.17}$$

This is a generalization of (A.12) and (A.13).

### A.4 Interpretation of the Weyl quantization and expansion of the star product

To see the origin of Weyl quantization procedure, it is useful to introduce the Fourier transform of the symbol

$$\tilde{f}(\eta,\xi) = \int \mathrm{d}y\mathrm{d}p \, f(y,p)e^{-\frac{i}{\epsilon}(y\eta+p\xi)}, \tag{A.18}$$

$$f(y,p) = \frac{1}{(2\pi\epsilon)^2} \int \mathrm{d}\eta\mathrm{d}\xi \, \tilde{f}(\eta,\xi) \, e^{\frac{i}{\epsilon}(y\eta+p\xi)}. \tag{A.19}$$

The operator $\hat{f}$ is recovered by replacing $p$ with $-i\epsilon\partial_y$ in this last expression, which allows for a direct interpretation of the Weyl transform as a quantization procedure:

$$\hat{f}(y,\partial_y) = \frac{1}{(2\pi\epsilon)^2} \int \mathrm{d}\eta\mathrm{d}\xi \, \tilde{f}(\eta,\xi) \, e^{\frac{i}{\epsilon}\eta y+\xi\partial_y}. \tag{A.20}$$

Expression (A.20) is sometimes used as a definition for the Weyl transform [49]. To check that this definition is equivalent to (A.3), recall two useful formula involving the exponential function of operator derivative $\partial_y$:

$$e^{i\frac{\eta}{\epsilon}y+\xi\partial_y} = e^{i\frac{\eta\xi}{2\epsilon}}e^{i\frac{\eta}{\epsilon}y}e^{\xi\partial_y}\,, \tag{A.21}$$

$$e^{\xi\partial_y}\psi(y) = \psi(y+\xi)\,. \tag{A.22}$$

The first equality is a particular case of Baker–Campbell–Hausdorff formula commonly encountered in physics [50].

The definition (A.20) of the Weyl transform is a useful one to derive the product rule (A.15) from the definition (A.14) of the star product, as shown for instance in Refs. [49,51], among others. Indeed, the operator product is conveniently written in terms of the Fourier transform of their symbol using (A.20), followed by (A.22):

$$\hat{f}\hat{g} = \frac{1}{(2\pi\epsilon)^4}\int d\eta d\xi d\eta' d\xi' \tilde{f}(\eta,\xi)\tilde{g}(\eta',\xi')e^{\frac{i}{\epsilon}\eta y+\xi\partial_y}e^{\frac{i}{\epsilon}\eta'y+\xi'\partial_y}\,, \tag{A.23}$$

$$= \frac{1}{(2\pi\epsilon)^4}\int d\eta d\xi d\eta' d\xi' \tilde{f}(\eta,\xi)\tilde{g}(\eta',\xi')e^{\frac{i}{2\epsilon}(\eta'\xi-\eta\xi')}e^{\frac{i}{\epsilon}(\eta+\eta')y+(\xi+\xi')\partial_y}\,, \tag{A.24}$$

$$= \frac{1}{(2\pi\epsilon)^2}\int d\eta'' d\xi'' \widetilde{f\star g}(\eta'',\xi'')e^{\frac{i}{\epsilon}\eta''y+\xi''\partial_y}\,, \tag{A.25}$$

where the last equality is just the definition of the Weyl transform (A.20) combined to (A.14). Identifying the last two lines leads to

$$\widetilde{f\star g}(\eta'',\xi'') = \frac{1}{(2\pi\epsilon)^2}\int d\eta d\xi d\eta' d\xi' \tilde{f}(\eta,\xi)\tilde{g}(\eta',\xi')e^{\frac{i}{2\epsilon}(\eta'\xi-\eta\xi')}\delta(\xi''-\xi'-\xi)\delta(\eta''-\eta'-\eta)\,. \tag{A.26}$$

The expression (A.15) for the star product $f\star g$ is recovered by inserting (A.26) in (A.19), using the expansion

$$e^{\frac{i}{2\epsilon}(\eta'\xi-\eta\xi')} = \sum_{(n,m)\in\mathbb{N}^2}\frac{(-1)^n}{n!m!}\left(\frac{i}{2}\epsilon\right)^{n+m}\left(\frac{i\eta}{\epsilon}\right)^m\left(\frac{i\xi}{\epsilon}\right)^n\left(\frac{i\eta'}{\epsilon}\right)^n\left(\frac{i\xi'}{\epsilon}\right)^m\,, \tag{A.27}$$

together with basic properties of inverse Fourier transforms.

## A.5 Symbolic calculus with a WKB ansatz

We recall classical results on the asymptotic development of an operator $\hat{f}$ with symbol $f(y,p)$ acting on a scalar field

$$\psi = a(y)e^{i\frac{\phi(y)}{\epsilon}}\,. \tag{A.28}$$

This is just a translation in our notations for the particular one-dimensional case. Using the definition (A.3) together with a change of variable $y'' = y'-y$ leads to

$$\hat{f}\psi(y) = \frac{1}{2\pi\epsilon}\int dy'' dp\, f\left(y+\frac{y''}{2},p\right)a(y+y'')e^{\frac{i}{\epsilon}(\phi(y+y'')-y''p)}\,. \tag{A.29}$$

A Taylor expansion of $f$, $a$ and $\phi$ in terms of $y''$ (that will be justified a posteriori) yields

$$\hat{f}\psi(y) = \frac{\psi(y)}{2\pi\epsilon}\int dy'' dp\left(f+y''\left(\frac{\partial_y f}{2}+\frac{f\partial_y a}{a}\right)+\mathcal{O}(y''^2)\right)e^{\frac{i}{\epsilon}\left((\partial_y\phi-p)y''+\frac{\partial_{yy}\phi}{2}y''^2+\mathcal{O}(y''^3)\right)}\,, \tag{A.30}$$

where all functions inside the integral are evaluated at $y$. The term in the exponential is expanded up to the second order because of the $1/\epsilon$ prefactor. Powers of $y''$ in the integrand can be replaced by derivatives with respect to $p$ in front of the exponential term. Keeping only terms up to order $\epsilon$ yields to:

$$\hat{f}\psi(y) = \frac{\psi(y)}{2\pi\epsilon}\int \mathrm{d}y''\mathrm{d}p\left(f + i\epsilon\left(\frac{\partial_y f}{2} + \frac{f\partial_y a}{a}\right)\partial_p - i\epsilon f\frac{\partial_{yy}\phi}{2}\partial_{pp} + \mathcal{O}(\epsilon^2)\right)e^{\frac{i}{\epsilon}(\partial_y\phi - p)y''}. \quad (A.31)$$

We now expand the symbol, the amplitude and phase functions as

$$f = f_0(y,p) + \epsilon f_1(y,p) + \mathcal{O}(\epsilon^2), \quad a = a_0(y) + \mathcal{O}(\epsilon), \quad \phi = \phi_0(y) + \epsilon\phi_1(y) + \mathcal{O}(\epsilon^2). \quad (A.32)$$

After integrations by parts for the variable $p$ in Eq (A.31), and after using the identity

$$\frac{1}{2\pi\epsilon}\int \mathrm{d}y''\, g(y,p)e^{\frac{i}{\epsilon}(\partial_y\phi - p)y''} = g(y,\partial_y\phi)\delta\left(\partial_y\phi - p\right), \quad (A.33)$$

with $\delta(x)$ the Dirac distribution, we find

$$\hat{f}\psi = f_0\psi + \epsilon\left(f_1 - \frac{i}{2}\left(\partial_{pp}f_0\partial_{yy}\phi_0 + \partial_{yp}f_0 + \partial_p f_0\partial_y \ln(a_0^2)\right)\right)\psi + \mathcal{O}(\epsilon^2), \quad (A.34)$$

where the symbols $f_0$ and $f_1$ are evaluated at $(y,p(y))$ with

$$p = \partial_y\phi_0 + \epsilon\partial_y\phi_1. \quad (A.35)$$

## B  Quantum harmonic oscillator limit

We show in this subsection that eigenmodes with sufficiently small $n$ index can be approximated by solutions of a differential equation analogous to the quantum harmonic oscillator problem [36], in the semi-classical limit $\epsilon \to 0$. This amounts to consider the limit of small radius $\varrho$ for trajectories in phase space, or equivalently, the limit of frequencies asymptotically close to one: $|\omega - 1| \ll 1$.

Before presenting the computation, let us stress that operators $\hat{\Omega}^{\pm}$ are not differential operators in general. By contrast, the original multicomponent wave operator $\underline{\hat{\mathcal{H}}}^{\pm}$ is a differential operator. Differential operators involve only terms such as $\partial^n/\partial y^n$ with $n$ a non-negative integer. Consequently, their symbols involve only polynomial terms in the wavenumber $p$. However, the scalar operators $\hat{\Omega}^{\pm}$ are obtained after a diagonalization procedure of the symbols $\underline{\mathcal{H}}^{\pm}$, and this diagonalization leads to the presence of non-polynomial terms in the expression of $\Omega^{\pm}(y,p)$, see for instance the r.h.s. of (70). Owing to the presence of those non-polynomial terms, the corresponding operators $\hat{\Omega}^{\pm}$ belongs to the family of *pseudo-differential operators*.

In the limit of trajectories that are close to the origin in phase space $(y,p)$, a Taylor expansion allows us to turn the pseudo-differential operators into differential operators analogous to the quantum harmonic oscillator operator, up to a constant. In the shallow water case, taking the limit $\varrho \ll 1$ in (70) leads to

$$\omega^{\pm} = 1 + \frac{1}{2}\varrho^2 \mp \frac{\epsilon}{2}\left(1 - \varrho^2\right) + \mathcal{O}(\varrho^4). \quad (B.1)$$

The term $\epsilon\varrho^2$ can also be dropped in the small $\epsilon$ limit. Then, using $\Omega^{\pm} = \omega^{\pm} + \mathcal{O}(\epsilon^2)$, $\varrho^2 = y^2 + p^2$, and applying Weyl quantization to this symbol leads to

$$\hat{\Omega}^{\pm} = \frac{1}{2}\left(-\epsilon^2\frac{d^2}{dy^2} + y^2\right) + 1 \mp \frac{\epsilon}{2} + \mathcal{O}(\epsilon^2), \quad (B.2)$$

The eigenvalues of this operator are $\omega^{\pm} = (m^{\pm} + 1/2)\epsilon + 1 \mp \epsilon/2$ with $m^{\pm} \in \mathbb{N}$ [36]. Now, we want to match this dispersion relation with the semi-classical result obtained in the main text under the condition $\varrho \gg \epsilon$, while keeping $\varrho \ll 1$ to stay in the harmonic oscillator limit. This is done by choosing $m^{\pm} = n \pm 1$:

$$\omega^{+} = 1 + \epsilon(n+1), \quad n \geq -1, \tag{B.3}$$

$$\omega^{-} = 1 + \epsilon n, \qquad n \geq 1. \tag{B.4}$$

The expression $n(\omega)$ is drawn as a dashed line in Fig. 5d.

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
