# Peer review of "From ray tracing to waves of topological origin in continuous media"

_SciPost Physics, doi:SciPost Phys. 14, 062 (2023)_

## Round 1 · Referee Report · Anonymous (Referee 1) · 2022-8-28

Strengths

1- devoted to the topical subject 2-modern methods are used for solving the problem 3- the solution fills the gap in previous similar discussions of the problem

Weaknesses

1-the paper is mainly methodical

Report

In this paper, the problem of prediction of existence of topological waves by the quasi-classical analysis of a corresponding non-uniform linear wave system is discussed for a
simple case of equatorial shallow water waves. It is shown that the difference of numbers of modes of linear wave operators in quasi-classical limit at $k\to\pm\infty$ gives the number of topological modes. The Bohr-Sommerfeld quantization rule includes Berry curvature contribution resulting in the above mentioned difference. Although this result is not essentially new, the method seems useful and effective enough for application to other
problems of modern wave physics. The paper is written very clearly and the exposition is very pedagogical, so any reader with basic knowledge of quantum mechanics in phase space can follow the text without much difficulty. I have only noticed a few misprints in the text. After their correction the paper can be published in SciPost.

Requested changes

1- page 14, top line: the symbol $\hat{H}^{\pm}$ should be replaced by $\hat{\mathcal{H}}^{\pm}$, 2- page 14, line 7 from bottom: number 54 should be replaced by (54), 3- page 14, line 3 from bottom: expect' should be replaced byexcept', 4- page 17, Eq.~(62): $\Omega$ should be replaced by $\Omega_s$, 5- page 18, line below Eq.~(66): (62) should be replaced by (64), 6- page 24, line below Eq.~(88): start' should be replaced bystar', 7- page 24, line 2 below Eq.~(92): $y$ should be replaced by $y^{\prime\prime}$, 8- page 27, in [48] number 54 should be replaced by (54).

---

## Round 1 · Referee Report · Brad Marston (Referee 2) · 2022-8-31

Strengths

  1. Provides a physical picture, rooted in a semiclassical approximation, for the still rather mysterious principle of bulk-interface correspondence that is widely used to understand waves of topological origin in fluids.

  2. Provides a starting point for future investigations of related systems such as non-Hermitian linear waves in fluids (due to forcing and dissipation).

  3. The semiclassical tools are likely applicable to a wide range of other systems (both classical and quantum) that support waves of topological origin.

  4. Makes interdisciplinary connections between several fields of physics (geophysical fluids, quantum mechanics, topological physics).

Thus the paper meets 3 if not 4 of the expected criteria for acceptance (with only 1 required for acceptance).

Weaknesses

  1. The paper is a challenging read due to the use of mathematical tools that will be unfamiliar to many readers. The need for such tools isn't completely clear to me.

  2. I suspect that a simpler but more accessible presentation (perhaps with some reduction of rigor) would be possible but this isn't addressed.

  3. See minor suggestions for changes listed below.

Report

Bulk-boundary correspondence lies at the heart of the topological origin of certain waves that propagate along the edges of various forms of matter. Roughly speaking it postulates that non-trivial topology of waves in the bulk far from edges implies the existence of waves localized to the edge at frequencies or energies that are forbidden in the bulk. The edge currents of the integer quantum Hall effect are a well-known example. Thouless, Kohmoto, Nightingale, and Dennijs showed in 1982 that the conductance was quantized because the (integer) number of edge modes followed from non-trivial topology of the bulk Landau levels as characterized by a non-zero Chern number.

Yet a physical understanding of the basis of bulk-boundary correspondence has remained elusive, despite the existence of some rigorous mathematical arguments backing up the notion. The authors of the present paper make an important step towards achieving such a physical understanding for the closely related concept of bulk-interface correspondence. The illustrative fluid system that they consider is the shallow water equations on a rotating sphere (planet). The equator is the interface between the two hemispheres where the Coriolis parameter changes sign. One of the authors (Venaille) along with Pierre Delplace and this reviewer showed in 2017 that equatorial Kelvin and Yanai waves previously identified by geophysicists have a topological origin. However the ray tracing procedure presented in the paper appears to be much more general and should be applicable both to condensed matter and other systems.

The authors introduce some mathematical tools including Weyl-Wigner transformations from operators to functions that are called “symbols” by mathematicians (which has an unfortunate mystical connotation) and Moyal star products that will be unfamiliar to many readers. These will present a barrier to those readers (fortunately an Appendix provides a quick tutorial while longer papers by Onuki and Littlejohn and Flynn provide background. But it seems possible to reformulate the calculations using more conventional tools. For example Section III A “Wave packet center of mass and wave packet momentum” seem rather standard. In any case the tools are combined with the semi-classical WKB approximation. By tracing the rays and carefully application of the quantization condition in the semiclassical limit (taking into account the contribution from the Berry monopole), spectral flow is recovered (and thus bulk-interface correspondence) for the shallow water system on a rotating sphere. This tour-de-force is illuminating because it shows how the non-trivial topology of the bulk fluid leads to modes along equator.

I recommend publication in SciPost.

Requested changes

  1. The first sentence of the abstract reads “Inhomogeneous media commonly support a discrete number of waves modes that are trapped along interfaces defined by spatially varying parameters.” The word “inhomogeneous” seems too general here as it also would include systems that are spatially random in all directions (eg. Anderson localization).

  2. The shorthand notation “topological waves” used in the paper while convenient, is a bit misleading because while the waves have an origin rooted in topology, they aren’t exactly topological themselves.

  3. Near the end of the introduction the spectral flow index is defined to be “an integer that counts the number of topological modes that transit from one wave band to another….” but more precisely shouldn’t this be the difference between the number of right movers and left movers (also in Eq. 6)? In other words if the spectral flow index = 2, could there not be 3 right moving waves along the interface and 1 left mover?

  • validity: high
  • significance: high
  • originality: high
  • clarity: high
  • formatting: excellent
  • grammar: excellent

Author:  Antoine Venaille  on 2022-10-11  [id 2912]

(in reply to Report 2 by Brad Marston on 2022-08-31)

1- Ok

2 - We agree that the term Wave of topological origin is better and should be used when possible. We have changed the title accordingly. We keep this convenient shorhand term in the text, but we added some warning on this terminology.

3 - We have added a paragraph in the section were spectral flow is defined:

\textit{Note that in many cases, the spectral flow is directly related to the difference between the number of right-moving (positive group velocity) and left-moving modes in a given range of frequencies, most often taken within the gap between different bulk wavebands when it exists. In Matsuno's case, at any nonzero frequency, there are two more modes with eastward group velocity than modes with westward group velocity, and this is related to the spectral flow index $\pm2$ of the upper and low wave bands [Delplace et al 2017]. Thus, since group velocity corresponds to energy transport, spectral flow is related to unidirectional wave transport in that case.}

Note that in many cases, the spectral flow index can be directly related to another index that counts the difference between right moving (positive group velocity) and left moving (negative group velocity) in a given range of frequencies, most often taken within the gap between different bulk wavebands when it exists. In Matsuno case, at any frequency, there are two more right-moving modes (positive group velocity) than left-moving modes (negative velocity), and this is related to the spectral flow index $\pm2$ of the upper and low wave bands. Thus, spectral flow is related to unidirectional wave transport.}

We did not want to go into more detailed, as there are situations where those two indices may be difference (for instance in cases of indirect gap between different wavebands), and the spectral flow index turns out to be more general (even if the other related to unidirectional propagation is admittedly more important for physical consequences of topology).}

---

## Round 1 · Referee Report · Anonymous (Referee 3) · 2022-9-1

Strengths

  1. The work brings new insights into topological analysis in continuous media and complements many previous works that analyze the topology of shallow water equations through bulk-boundary/interface correspondence (e.g., Delplace et al. Science 2017).
  2. Instead of relying on the f-plane approximation, the authors use the beta-plane approximation in which there is a spatial dependence, and the bulk-interface shows up naturally.
  3. It provides some physical interpretation for the bulk-interface correspondence.
  4. Overall, the presentation is very clear and the paper is very pedagogical.

Weaknesses

  1. There is a lack of comparison with previously published work on the shallow water system with an f-plane approximation.
  2. The Wigner-Weyl transform of the shallow water equations is not given explicitly.
  3. There is a lack of rigorous connection between the quantization rule that the authors derived in Eq. (72) and the bulk-interface correspondence. I will elaborate on these points further later in the report (see #2, #9, #11 in the requested changes)

Report

In this work, the authors apply ray tracing machinery to the topological waves in shallow water equations. They show that Berry curvature arises naturally from the ray tracing equations and that the integral of Berry curvature is quantized, which agrees with the bulk-interface correspondence shown in previous work. Overall, this work brings new insights into topological analysis in classical continuous media and complements previous work. As such, I recommend publication with my detailed comments in the requested changes below.

Requested changes

  1. On page 8, the authors state that “$\hat{H}$ is a linear operator involving spatially varying coefficients and spatial derivatives, together with a parameter to be varied.” Could the authors clarify what is the parameter to be varied?

  2. On Page 11, how do I understand the definitions of $y_v$ and $p_v$ in Eq. (30)? In the quantum mechanical context, these are just the expectation values of the position and momentum operators, but here because $\Psi$ is the vector that consists of velocity field $(u, v, \eta)$, the interpretation is less clear to me. Is Eq. (31) just the scalar version of this? Can we understand $\psi$ as the scalar component of the $\Psi$?

  3. What is the normalization condition for $\psi$ in Eq. (34)?

  4. From Eq. (37) to Eq. (38), where is $a_0^2 (y)$? Is this integrating over y, and let $\int a_0^2 (y) dy = 1$ because $a_0(y)$ Is on the order of 1?

  5. It is not obvious to me how the symbolic order 0 Hamiltonian in Eq. (25) is derived. In Appendix A, the authors define the Wigner-Weyl transform, which requires the knowledge of F(y,y’). In general, how do we find F(y,y’) for a given operator? How do the authors derive the symbol Hamiltonian Eq. (54) from Eq. (25) using the Wigner-Weyl transform? Could this be given explicitly?

  6. How is the gauge-invariant problem in Eq. (45)? It seems to not be used in Eqs. (46)-(48).

  7. On Page 16, the authors state that the Chern number “describes how twisted is the whole bundle of eigenvectors.” However, the Chern number is a global quantity, but whether the bundle is twisted sounds like a local description. Also, doesn’t the Chern number simply describe the number of phase singularities? For example, if C = 1, it simply means that there is one phase singularity, and the wavevector can only have divergence and no twist at all (in other words, the curl could be 0).

  8. On Page 17, the authors mention the Berry monopole — this could benefit from a little more discussion about the Berry monopole for the fluid dynamics audience.

  9. How is this work related to the methodology in the previous work by Delplace et al. Science 2017, in which they use the f-plane approximation to calculate the Chern number directly?

  10. To use the f-plane approximation to obtain the Chern number, people often use the odd viscosity to ensure convergence at large k. Does the ray tracing method bypass the need for regularizing the solution at large k’s?

  11. The authors derive Eq. (72) from ray tracing and make the argument that the number of unpaired modes is 2, which agrees with the expectation from the bulk-interface correspondence. How do we know the n=-1 and n = 0 modes are the interfacial modes (Rossby modes, for example, are localized at the equator too)? Do we know that they are topologically protected? Can we make a rigorous connection with the bulk-interface correspondence?

  12. Could the authors comment on the generalizability of Wigner-Weyl transformation? Could this be applied to other similar classes of problems such as the coastal Kelvin waves, shallow water equations with a shear flow, primitive equations, or other continuous media?

  • validity: top
  • significance: high
  • originality: good
  • clarity: top
  • formatting: excellent
  • grammar: excellent

Author:  Antoine Venaille  on 2022-10-11  [id 2908]

(in reply to Report 3 on 2022-09-01)

Thanks for your carefull reading of the manuscript and suggestions.

1- We changed the text as follows

\textit{together with an external parameter $k$ to be varied. In the shallow water case, this \textit{spectral flow parameter} $k$ is the wavenumber in zonal (West-East) direction.}

2 - This is an important point. Indeed, because of the vectorial nature of the state vector, the interpretation of the averaged quantities $y_v$ abd $p_v$ is not straightforward. It becomes clear when noting that it corresponds to the mean position and mean momentum averaged spatially with a weight given by the local energy (kinetic + potential ). We have added the following remark following the definition of $y_v$ and $p_v$:

\textit{ In the quantum mechanical context, $y_v$ and $p_v$ are just the expectation values of the position and momentum operators. Recall that the normalization constraint (10) is equivalent to the energy conservation for shallow water waves. This is why $y_v$ and $p_v$ are respectively interpreted in this context as an averaged energy-weighted position and momentum (wavenumber) for a given wavepacket. The weight $\Psi^2$ corresponds indeed to the sum of local kinetic and potential energy, which, in dimensional units, is $0.5(u^2+v^2+g\eta^2)$}.

Then, after introducing $y_s$ and $p_s$, we now add the following remark:

\textit{The quantities $(y_s,p_s)$ in (31) are averaged position and momentum, just as $(y_v,p_v)$ in (30), albeit with a different weight. In the case of $(y_v,p_v)$, the weight was the local energy density, a physically meaningful quantity. This is not the case for $(y_s,p_s)$. In fact, we will see in subsection IIIC that $(y_s,p_s)$ are not physical observable since they depend on the gauge choice for the wave vector reconstruction operator $\hat{\underline{\chi}}$ that relates the scalar field $\psi$ to the vector field $\underline{\Psi}$ through (9).}

3- We have added the following sentence

\textit{The normalisation condition (10) for $\psi$ leads to the constraint} \begin{equation} { \int \mathrm{d} y \ a_0^2 =1 .} \end{equation}

4 - Yes. We now refer explicitly to the normalisation condition for $a_0$, introduced thanks to the answer to your third point

5 - We have developed the appendix on Wigner Weyl transform, in an attempt to be more pedagogical. (a) In particular, we now explain that it is not necessary to know the Kernel $F(y,y’)$ to compute the symbol associated with a given operator in most practical situations as in this paper. We provide in a new subsection of the appendix several simple but useful explicite results on the symbol-operator correspondence, and explain how to obtain more general results by using product formula. The simple cases are sufficient to obtain the symbol of the shallow water wave operator. We also now give an alternative (but equivalent) formula for the Weyl transform that is useful to derive the Moyal product formula.

6 - We think their was a confusion because we did not write explicitly the dependence of $\Omega_s$ on $(y_s,p_s)$ in Eq. (45) (Eq. (46) in new version), and because we forgot to say in previous version that this expression must also be used to derived the new set of equation. We have also added the following remark:

\textit{In particular, the term $\Omega_{1B}$ present in (46) has been cancelled out in (49) by a contribution induced by the change of variable from $(y_s,p_s)$ to $(y_v,p_v)$. This was expected as we already noticed that $y_v$ and $p_v$ are physical observable interpreted as averaged position and momentum with a local energy density weight, and the temporal evolution of such physical observables can not be gauge-dependent.}

7 - We agree that the Chern number is a global quantity, and the sentence mentioned by the referee was indeed confusing, so we removed it. We also modified the introduction of this subsection to be clear that Berry curvature is a geometrical quantity describing local twisting while the first Chern number counts singularities by integration of the Berrry curvature. We also slightly modified to paragraph interpreting Chern-Gauss-Bonnet formula. About the last statement of the referee, may be some confusion comes from the figure showing the Berry curvature vector. This field is curl-free, but the bundle of eigenvector (the object of interest here) is not curl-free whenever the Chern is not zero. We added a remark in the caption to avoid this confusion.

8 - We have rewritten the paragraph entitled "physical interpretation…" to give more explanation on the origin of the term Berry monopole:

\textit{ \textbf{Physical interpretation: analogy with a magnetic monopole.} As noted previously, the Berry curvature does not depend on the phase choice for the normalized eigenvector $\underline{\chi}$. It describes how fast the polarization relation changes when parameters are varied in the vicinity of a given point in parameter space. A direct consequence of (61) together with the existence of a degeneracy point carrying a non-zero Chern number is that the Berry curvature diverges close to the band-degeneracy points. In fact, one can interpret this curvature $\mathbf{F}$ as being generated by a \textit{Berry monopole}, in the same way as a divergence or convergent magnetic field would be generated by a positive or negative magnetic monopole, whose charge must be quantized, according to a celebrated work by Dirac [Nakahara 2018, Delplace 2021]. In this analogy the first Chern number plays the role of the magnetic charge of the monopole.}

9 - In Delplace et al, the same Chern numbers were computed. Thus, this previous paper already computed the topological properties of the symbol considered here. It was also noticed in Delplace et al 2017 that this number was equal to the equatorial spectral flow. Our contribution here is to explain the physical origine of this equivalence between a Chern number and a spectral flow, through the use of ray tracing equation that involve the same f-plane symbol as in Delplace et al 2017. With the ray tracing formalism, f becomes a slowly varying parameter instead of being an externally prescribed parameter. This allows to connect the topological index to the spectral index, after proper use of quantification relation. We clarified this important point in the new version of the conclusion (first part).

10 - The short answer is that ray tracing does not bypass the need for a regularization procedure if one wants to show the topological origin of coastal waves on the f plane with sharp walls. We added a whole paragraph in the conclusion addressing this important point.

\textit{The \emph{bulk-interface} correspondence discussed in this paper is different than the \emph{bulk-boundary} correspondence that is commonly encountered in condensed matter. In that case, a bulk Chern number can be defined on each sides of the interface (e.g. each side of the equator for shallow water waves). Assigning a topological invariant on each side of the interface for continuous media is not always possible. For instance, for rotating shallow water waves, it requires the introduction of a regularization parameter such as odd-viscosity [Volovik 1988, Bal 2019, Souslov et al 2019, Tauber etal 2019]. When this parameter is added into the model, one can assign a topological index to the $f$-plane problem, i.e. in each hemisphere, and predict the number of unidirectional modes at the equator [Tauber et al 2019] The case of sharp (discontinuous) interfaces or hard boundaries such as solid walls is much more complicated for continuous media: unidirectional modes exist [Souslov et al 2019] yet apparent violation of standard bulk-boundary correspondence have been noted and explained [Tauber et al 2020, Graf et al 2021]. Our present work applies to the unbounded case and as such provide an explanation for the bulk-interface correspondence only. However, hard-boundary cases may in some cases be interpreted as limiting cases of an interfaces. For instance, unidirectional trapped modes along coasts found by Kelvin in 1880 [Thompson 2021] can be recovered from the study of a shallow water with varying bottom topography defining an interface at the coast [Venaille Delplace 2021]. In that context, the ray tracing approach may help to bridge the gap between the modern topological point of view and the more traditional textbook skipping orbit picture explaining the emergence of unidirectional trapped modes. It will also be interesting to relate this approach to the interpretation of rotating shallow water dynamics as a Chern-Simons topological field theory [Tong 2022].}

11 - In this problem, as in the normal form considered by Matsuno, all the modes are somewhat localized at the equator, as they are described by parabolic cylinder functions. The $n=-1$ and $n=0$ modes are more localized than the others because their number of zeros in the function $u$ is smaller. In phase space, the $n=-1$ and $n=0$ are more localized than the others as their trajectories is closed to the origin. Those modes remain trapped as $f(y)$ is varied from the beta plane to a step function, while the other modes would become delocalized, as shown for instance in [Tauber et al JFM 2019]. It would be interesting to describe this with the ray tracing approach, in a future work.

If by topological protection the referee means robustness of the spectral flow under continuous deformation of the problem parameter (say the profile $f(y)$), then our result indeed explain the origin of this property. In the previous version, we omitted to mention an key point to reach this conclusion: the conservation of wave branches in the dispersion relation as k is varied, that allows us connect the semi-classical results (mode number imbalance in the limits $k\rightarrow\pm\infty$ ) to the spectral flow (net number of modes that transit from one waveband to another). This point is now given in the conclusion section:

\textit{Once trajectories are computed, the second key step is to use the Bohr-Sommerfeld selection rule to find the discrete set of wave-packet trajectories that correspond to eigenmodes of the equatorial beta-plane shallow water wave problem, and to compare the sets of eigenmodes in the limit $k\rightarrow+\infty$ and $k\rightarrow-\infty$, for the upper frequency Poincar\'e wave band. Any mismatch is interpreted as a spectral flow for varying $k$. Indeed, wave branches in the dispersion relation can neither be created nor anihilated as $k$ is varied. The reason is that the wave operator is self-adjoint for a given $k$. As such, its spectrum defines a complete orthonormal basis for triplet of 1D fields in $y$ direction. Thus, if a wave-band has more modes for $k\rightarrow+\infty$ than for $k\rightarrow-\infty$, it means that the additional modes have transited from other wave-bands as $k$ was varied.}

Rigorous results on index theorem relating topological properties such as those computed here to spectral flows already exist, as mentioned in the text (work by Faure, Bal, Tauber, Delplace, and others). We do not claim to give any new results at a mathematical level here. Instead, we hope to provide through formal ray tracing computation some new physical intuition on this rather abstract bulk-interface correspondence.

12 - The approach can be generalized to any continuous media problem with Hermitian wave operator. We have added a paragraph in the conclusion

\textit{ We stress that we focused on a semi-classical interpretation of a \emph{bulk-interface} correspondence, in problems without boundaries. In such problems, waves of topological origin emerge along an interface of a parameter that opens a frequency gap for the bulk waves, as the Coriolis parameter for equatorial waves. While the method was presented in the case of equatorial shallow water waves, it is sufficiently general to be applied to a much wider class of problems including plasma, continuously stratified and compressible flows, among others. In all those problems with Hermitian wave operator, topological features are found by looking for Berry monopole in parameter space, and we have shown how such monopole may affect wavepacket dynamics in phase space. }

The non-Hermitian case is less clear, but certainly worth studying in future work, and we developed a paragraph on this point at the end of the conclusion:

\textit{ It will also be interesting to see in future work how non-Hermitian wave problem can be tackled with the ray tracing approach. Such problems naturally occur in the context of flow instabilities [Smyth Carpenter 2019], or in the presence of dissipative terms in the dynamics. On the one hand, some of the spectral flow properties found in the Hermitian case seem to be robust to the presence of non-Hermitian effects, as reported in the context of shallow water dynamics in the presence of a mean velocity shear [Zhu et al 2022]. On the other hand, intriguing new unidirectional trapped modes have been reported in rotating convection [Favier et al 2020]. The ray tracing point of view may help to bring new hindsight on the topological origin of those phenomena.}

---

## Round 2 · Referee Report · Anonymous (Referee 1) · 2022-11-4

Report

The exposition in the revised version is considerably improved. In particular, all my corrections are taken into account. I recommend publication of this paper as it is.

---

## Round 2 · Referee Report · Brad Marston (Referee 2) · 2022-11-27

Report

I am satisfied by the response of the authors to my report, and the comments and questions of reviewer #3. There are no further requested changes, and I strongly recommend acceptance of this important paper for publication in SciPost.

---

## Round 2 · Referee Report · Anonymous (Referee 3) · 2022-12-2

Strengths

Clear presentation and bringing new perspectives into understanding topological properties of continuous media.

Report

The authors have addressed my point in a satisfactory manner. They provided many clarifications and made the introduction to the Wigner-Weyl transformation in the appendix more pedagogical. Therefore, I recommend publication in SciPost.

---

## Round 2 · Author Response

Dear Editor,

We thank the reviewers for carefully reading our preprint, for their useful comments and suggestions to improve presentation of the paper.

We addressed the requested changes of all reviewers (see below). Reviewer 2 wondered whether a less technical explanation of the result could be provided, avoiding jargon from symbolic calculus. We have writen a new discussion section that is much longer (3 pages) than in the first version, to give the main ideas of the paper without the use of technical terms. We now also provide more details in the appendix tutorial op Weyl calculus. Reviewer 3 asked for more connection with previous work on the shallow water or similar flow model in the context of topological waves, and asked for clarification on our results and the concept of bulk-boundary correspondence that is commonly used in physics. Those points are now also addressed in the conclusion section.

All the modifications brought to the previous version are emphasized in red in a new "highlighted" version that we can provide if needed.

Best regards

---

## Round 2 · List of Changes

\section{Requests from reviewer 1}

We have corrected all the typos found by the reviewer.

\section{Requests from reviewer 2}

\begin{enumerate} \item The first sentence of the abstract reads “Inhomogeneous media commonly support a discrete number of waves modes that are trapped along interfaces defined by spatially varying parameters.” The word “inhomogeneous” seems too general here as it also would include systems that are spatially random in all directions (eg. Anderson localization).

{\color{blue}
Ok}

\item The shorthand notation “topological waves” used in the paper while convenient, is a bit misleading because while the waves have an origin rooted in topology, they aren’t exactly topological themselves.

{\color{blue}
We agree that the term Wave of topological origin is better and should be used when possible. We have changed the title accordingly. We keep this convenient shorhand term in the text, but we added some warning on this terminology. }

\item Near the end of the introduction the spectral flow index is defined to be “an integer that counts the number of topological modes that transit from one wave band to another….” but more precisely shouldn’t this be the difference between the number of right movers and left movers (also in Eq. 6)? In other words if the spectral flow index = 2, could there not be 3 right moving waves along the interface and 1 left mover?

{\color{blue} We have added a paragraph in the section were spectral flow is defined:

\textit{Note that in many cases, the spectral flow is directly related to the difference between the number of right-moving (positive group velocity) and left-moving modes in a given range of frequencies, most often taken within the gap between different bulk wavebands when it exists. In Matsuno's case, at any nonzero frequency, there are two more modes with eastward group velocity than modes with westward group velocity, and this is related to the spectral flow index $\pm2$ of the upper and low wave bands [Delplace et al 2017]. Thus, since group velocity corresponds to energy transport, spectral flow is related to unidirectional wave transport in that case.}

Note that in many cases, the spectral flow index can be directly related to another index that counts the difference between right moving (positive group velocity) and left moving (negative group velocity) in a given range of frequencies, most often taken within the gap between different bulk wavebands when it exists. In Matsuno case, at any frequency, there are two more right-moving modes (positive group velocity) than left-moving modes (negative velocity), and this is related to the spectral flow index $\pm2$ of the upper and low wave bands. Thus, spectral flow is related to unidirectional wave transport.}

We did not want to go into more detailed, as there are situations where those two indices may be difference (for instance in cases of indirect gap between different wavebands), and the spectral flow index turns out to be more general (even if the other related to unidirectional propagation is admittedly more important for physical consequences of topology).}

\end{enumerate}

\section{Requests from reviewer 3}

\begin{enumerate} \item On page 8, the authors state that $hat H$ is a linear operator involving spatially varying coefficients and spatial derivatives, together with a parameter to be varied.” Could the authors clarify what is the parameter to be varied?

{\color{blue} We changed the text as follows

\textit{together with an external parameter $k$ to be varied. In the shallow water case, this \textit{spectral flow parameter} $k$ is the wavenumber in zonal (West-East) direction.} }

\item On Page 11, how do I understand the definitions of $y_v$ and $p_v$ in Eq. (30)? In the quantum mechanical context, these are just the expectation values of the position and momentum operators, but here because $\underline{\Psi}$ is the vector that consists of velocity field $(u,v,\eta)$, the interpretation is less clear to me. Is Eq. (31) just the scalar version of this? Can we understand $\psi$ as the scalar component of the $\underline{\Psi}$?

{\color{blue} This is an important point. Indeed, because of the vectorial nature of the state vector, the interpretation of the averaged quantities $y_v$ abd $p_v$ is not straightforward. It becomes clear when noting that it corresponds to the mean position and mean momentum averaged spatially with a weight given by the local energy (kinetic + potential ). We have added the following remark following the definition of $y_v$ and $p_v$:

\textit{ In the quantum mechanical context, $y_v$ and $p_v$ are just the expectation values of the position and momentum operators. Recall that the normalization constraint (10) is equivalent to the energy conservation for shallow water waves. This is why $y_v$ and $p_v$ are respectively interpreted in this context as an averaged energy-weighted position and momentum (wavenumber) for a given wavepacket. The weight $\Psi^2$ corresponds indeed to the sum of local kinetic and potential energy, which, in dimensional units, is $0.5(u^2+v^2+g\eta^2)$}.

Then, after introducing $y_s$ and $p_s$, we now add the following remark:

\textit{The quantities $(y_s,p_s)$ in (31) are averaged position and momentum, just as $(y_v,p_v)$ in (30), albeit with a different weight. In the case of $(y_v,p_v)$, the weight was the local energy density, a physically meaningful quantity. This is not the case for $(y_s,p_s)$. In fact, we will see in subsection IIIC that $(y_s,p_s)$ are not physical observable since they depend on the gauge choice for the wave vector reconstruction operator $\hat{\underline{\chi}}$ that relates the scalar field $\psi$ to the vector field $\underline{\Psi}$ through (9).}

}

\item What is the normalization condition for $\psi$ in Eq. (34)?

{\color{blue} We have added the following sentence

\textit{The normalisation condition (10) for $\psi$ leads to the constraint} \begin{equation} { \color{blue} \int \mathrm{d} y \ a_0^2 =1 .} \end{equation}. }

\item From Eq. (37) to Eq. (38), where is $a_0^2(y)$? Is this integrating over $y$, and let $\int a_0^2(y)dy=1$ because $a_0(y)$ Is on the order of $1$? {\color{blue} Yes. We now refer explicitly to the normalisation condition for $a_0$, introduced thanks to the answer to your third point }

\item It is not obvious to me how the symbolic order 0 Hamiltonian in Eq. (25) is derived. (a) In Appendix A, the authors define the Wigner-Weyl transform, which requires the knowledge of $F(y,y’)$. In general, how do we find F(y,y’) for a given operator? (b) How do the authors derive the symbol Hamiltonian Eq. (54) from Eq. (25) using the Wigner-Weyl transform? Could this be given explicitly?

{\color{blue} We have developed the appendix on Wigner Weyl transform, in an attempt to be more pedagogical. (a) In particular, we now explain that it is not necessary to know the Kernel $F(y,y’)$ to compute the symbol associated with a given operator in most practical situations as in this paper. We provide in a new subsection of the appendix several simple but useful explicite results on the symbol-operator correspondence, and explain how to obtain more general results by using product formula. The simple cases are sufficient to obtain the symbol of the shallow water wave operator. We also now give an alternative (but equivalent) formula for the Weyl transform that is useful to derive the Moyal product formula. }

\item How is the gauge-invariant problem in Eq. (45)? It seems to not be used in Eqs. (46)-(48).

{\color{blue} We think their was a confusion because we did not write explicitly the dependence of $\Omega_s$ on $(y_s,p_s)$ in Eq. (45) (Eq. (46) in new version), and because we forgot to say in previous version that this expression must also be used to derived the new set of equation. We have also added the following remark:

\textit{In particular, the term $\Omega_{1B}$ present in (46) has been cancelled out in (49) by a contribution induced by the change of variable from $(y_s,p_s)$ to $(y_v,p_v)$. This was expected as we already noticed that $y_v$ and $p_v$ are physical observable interpreted as averaged position and momentum with a local energy density weight, and the temporal evolution of such physical observables can not be gauge-dependent.}

}

\item On Page 16, the authors state that the Chern number “describes how twisted is the whole bundle of eigenvectors.” However, the Chern number is a global quantity, but whether the bundle is twisted sounds like a local description. Also, doesn’t the Chern number simply describe the number of phase singularities? For example, if C = 1, it simply means that there is one phase singularity, and the wavevector can only have divergence and no twist at all (in other words, the curl could be 0).

{\color{blue} We agree that the Chern number is a global quantity, and the sentence mentioned by the referee was indeed confusing, so we removed it. We also modified the introduction of this subsection to be clear that Berry curvature is a geometrical quantity describing local twisting while the first Chern number counts singularities by integration of the Berrry curvature. We also slightly modified to paragraph interpreting Chern-Gauss-Bonnet formula. About the last statement of the referee, may be some confusion comes from the figure showing the Berry curvature vector. This field is curl-free, but the bundle of eigenvector (the object of interest here) is not curl-free whenever the Chern is not zero. We added a remark in the caption to avoid this confusion. }

\item On Page 17, the authors mention the Berry monopole — this could benefit from a little more discussion about the Berry monopole for the fluid dynamics audience.

{\color{blue} We have rewritten the paragraph entitled "physical interpretation…" to give more explanation on the origin of the term Berry monopole:

\textit{ \textbf{Physical interpretation: analogy with a magnetic monopole.} As noted previously, the Berry curvature does not depend on the phase choice for the normalized eigenvector $\underline{\chi}$. It describes how fast the polarization relation changes when parameters are varied in the vicinity of a given point in parameter space. A direct consequence of (61) together with the existence of a degeneracy point carrying a non-zero Chern number is that the Berry curvature diverges close to the band-degeneracy points. In fact, one can interpret this curvature $\mathbf{F}$ as being generated by a \textit{Berry monopole}, in the same way as a divergence or convergent magnetic field would be generated by a positive or negative magnetic monopole, whose charge must be quantized, according to a celebrated work by Dirac [Nakahara 2018, Delplace 2021]. In this analogy the first Chern number plays the role of the magnetic charge of the monopole.}

} \item How is this work related to the methodology in the previous work by Delplace et al. Science 2017, in which they use the f-plane approximation to calculate the Chern number directly? {\color{blue} 1. In Delplace et al, the same Chern numbers were computed. Thus, this previous paper already computed the topological properties of the symbol considered here. It was also noticed in Delplace et al 2017 that this number was equal to the equatorial spectral flow. Our contribution here is to explain the physical origine of this equivalence between a Chern number and a spectral flow, through the use of ray tracing equation that involve the same f-plane symbol as in Delplace et al 2017. With the ray tracing formalism, f becomes a slowly varying parameter instead of being an externally prescribed parameter. This allows to connect the topological index to the spectral index, after proper use of quantification relation. We clarified this important point in the new version of the conclusion (first part). } \item To use the f-plane approximation to obtain the Chern number, people often use the odd viscosity to ensure convergence at large k. Does the ray tracing method bypass the need for regularizing the solution at large k’s?

{\color{blue} The short answer is that ray tracing does not bypass the need for a regularization procedure if one wants to show the topological origin of coastal waves on the f plane with sharp walls. We added a whole paragraph in the conclusion addressing this important point.

\textit{The \emph{bulk-interface} correspondence discussed in this paper is different than the \emph{bulk-boundary} correspondence that is commonly encountered in condensed matter. In that case, a bulk Chern number can be defined on each sides of the interface (e.g. each side of the equator for shallow water waves). Assigning a topological invariant on each side of the interface for continuous media is not always possible. For instance, for rotating shallow water waves, it requires the introduction of a regularization parameter such as odd-viscosity [Volovik 1988, Bal 2019, Souslov et al 2019, Tauber etal 2019]. When this parameter is added into the model, one can assign a topological index to the $f$-plane problem, i.e. in each hemisphere, and predict the number of unidirectional modes at the equator [Tauber et al 2019] The case of sharp (discontinuous) interfaces or hard boundaries such as solid walls is much more complicated for continuous media: unidirectional modes exist [Souslov et al 2019] yet apparent violation of standard bulk-boundary correspondence have been noted and explained [Tauber et al 2020, Graf et al 2021]. Our present work applies to the unbounded case and as such provide an explanation for the bulk-interface correspondence only. However, hard-boundary cases may in some cases be interpreted as limiting cases of an interfaces. For instance, unidirectional trapped modes along coasts found by Kelvin in 1880 [Thompson 2021] can be recovered from the study of a shallow water with varying bottom topography defining an interface at the coast [Venaille Delplace 2021]. In that context, the ray tracing approach may help to bridge the gap between the modern topological point of view and the more traditional textbook skipping orbit picture explaining the emergence of unidirectional trapped modes. It will also be interesting to relate this approach to the interpretation of rotating shallow water dynamics as a Chern-Simons topological field theory [Tong 2022]. } }

\item The authors derive Eq. (72) from ray tracing and make the argument that the number of unpaired modes is 2, which agrees with the expectation from the bulk-interface correspondence. How do we know the n=-1 and n = 0 modes are the interfacial modes (Rossby modes, for example, are localized at the equator too)? Do we know that they are topologically protected? Can we make a rigorous connection with the bulk-interface correspondence? {\color{blue} In this problem, as in the normal form considered by Matsuno, all the modes are somewhat localized at the equator, as they are described by parabolic cylinder functions. The $n=-1$ and $n=0$ modes are more localized than the others because their number of zeros in the function $u$ is smaller. In phase space, the $n=-1$ and $n=0$ are more localized than the others as their trajectories is closed to the origin. Those modes remain trapped as $f(y)$ is varied from the beta plane to a step function, while the other modes would become delocalized, as shown for instance in [Tauber et al JFM 2019]. It would be interesting to describe this with the ray tracing approach, in a future work.

If by topological protection the referee means robustness of the spectral flow under continuous deformation of the problem parameter (say the profile $f(y)$), then our result indeed explain the origin of this property. In the previous version, we omitted to mention an key point to reach this conclusion: the conservation of wave branches in the dispersion relation as k is varied, that allows us connect the semi-classical results (mode number imbalance in the limits $k\rightarrow\pm\infty$ ) to the spectral flow (net number of modes that transit from one waveband to another). This point is now given in the conclusion section:

\textit{Once trajectories are computed, the second key step is to use the Bohr-Sommerfeld selection rule to find the discrete set of wave-packet trajectories that correspond to eigenmodes of the equatorial beta-plane shallow water wave problem, and to compare the sets of eigenmodes in the limit $k\rightarrow+\infty$ and $k\rightarrow-\infty$, for the upper frequency Poincar\'e wave band. Any mismatch is interpreted as a spectral flow for varying $k$. Indeed, wave branches in the dispersion relation can neither be created nor anihilated as $k$ is varied. The reason is that the wave operator is self-adjoint for a given $k$. As such, its spectrum defines a complete orthonormal basis for triplet of 1D fields in $y$ direction. Thus, if a wave-band has more modes for $k\rightarrow+\infty$ than for $k\rightarrow-\infty$, it means that the additional modes have transited from other wave-bands as $k$ was varied.}

Rigorous results on index theorem relating topological properties such as those computed here to spectral flows already exist, as mentioned in the text (work by Faure, Bal, Tauber, Delplace, and others). We do not claim to give any new results at a mathematical level here. Instead, we hope to provide through formal ray tracing computation some new physical intuition on this rather abstract bulk-interface correspondence.

} \item Could the authors comment on the generalizability of Wigner-Weyl transformation? Could this be applied to other similar classes of problems such as the coastal Kelvin waves, shallow water equations with a shear flow, primitive equations, or other continuous media?

{\color{blue} The approach can be generalized to any continuous media problem with Hermitian wave operator. We have added a paragraph in the conclusion

\textit{ We stress that we focused on a semi-classical interpretation of a \emph{bulk-interface} correspondence, in problems without boundaries. In such problems, waves of topological origin emerge along an interface of a parameter that opens a frequency gap for the bulk waves, as the Coriolis parameter for equatorial waves. While the method was presented in the case of equatorial shallow water waves, it is sufficiently general to be applied to a much wider class of problems including plasma, continuously stratified and compressible flows, among others. In all those problems with Hermitian wave operator, topological features are found by looking for Berry monopole in parameter space, and we have shown how such monopole may affect wavepacket dynamics in phase space. }

The non-Hermitian case is less clear, but certainly worth studying in future work, and we developed a paragraph on this point at the end of the conclusion:

\textit{ It will also be interesting to see in future work how non-Hermitian wave problem can be tackled with the ray tracing approach. Such problems naturally occur in the context of flow instabilities [Smyth Carpenter 2019], or in the presence of dissipative terms in the dynamics. On the one hand, some of the spectral flow properties found in the Hermitian case seem to be robust to the presence of non-Hermitian effects, as reported in the context of shallow water dynamics in the presence of a mean velocity shear [Zhu et al 2022]. On the other hand, intriguing new unidirectional trapped modes have been reported in rotating convection [Favier et al 2020]. The ray tracing point of view may help to bring new hindsight on the topological origin of those phenomena.}

} \end{enumerate}

---

## Editorial Decision

published